# Do Deep Generative Models Know
# What They Don't Know?

**Eric Nalisnick**[*][†], **Akihiro Matsukawa, Yee Whye Teh, Dilan Gorur, Balaji Lakshminarayanan**[*]
DeepMind

## Abstract

A neural network deployed in the wild may be asked to make predictions for inputs that were drawn from a different distribution than that of the training data. A plethora of work has demonstrated that it is easy to find or synthesize inputs for which a neural network is highly confident yet wrong. Generative models are widely viewed to be robust to such mistaken confidence as modeling the density of the input features can be used to detect novel, out-of-distribution inputs. In this paper we challenge this assumption. We find that the density learned by flow-based models, VAEs, and PixelCNNs cannot distinguish images of common objects such as dogs, trucks, and horses (i.e. CIFAR-10) from those of house numbers (i.e. SVHN), assigning a higher likelihood to the latter when the model is trained on the former. Moreover, we find evidence of this phenomenon when pairing several popular image data sets: FashionMNIST vs MNIST, CelebA vs SVHN, ImageNet vs CIFAR-10 / CIFAR-100 / SVHN. To investigate this curious behavior, we focus analysis on flow-based generative models in particular since they are trained and evaluated via the exact marginal likelihood. We find such behavior persists even when we restrict the flows to constant-volume transformations. These transformations admit some theoretical analysis, and we show that the difference in likelihoods can be explained by the location and variances of the data and the model curvature. Our results caution against using the density estimates from deep generative models to identify inputs similar to the training distribution until their behavior for out-of-distribution inputs is better understood.

## 1 Introduction

Deep learning has achieved impressive success in applications for which the goal is to model a conditional distribution $p(y|x)$, with $y$ being a label and $x$ the features. While the conditional model $p(y|x)$ may be highly accurate on inputs $x$ sampled from the training distribution, there are no guarantees that the model will work well on $x$'s drawn from some other distribution. For example, Louizos & Welling (2017) show that simply rotating an MNIST digit can make a neural network predict another class with high confidence (see their Figure 1a). Ostensibly, one way to avoid such overconfidently wrong predictions would be to train a density model $p(x; \theta)$ (with $\theta$ denoting the parameters) to approximate the true distribution of training inputs $p^*(x)$ and refuse to make a prediction for any $x$ that has a sufficiently low density under $p(x; \theta)$. The intuition is that the discriminative model $p(y|x)$ likely did not observe enough samples in that region to make a reliable decision for those inputs. This idea has been proposed by various papers, cf. (Bishop, 1994), and as recently as in the panel discussion at Advances in Approximate Bayesian Inference (AABI) 2017 (Blei et al., 2017).

Anomaly detection is just one motivating example for which we require accurate densities, and others include information regularization (Szummer & Jaakkola, 2003), open set recognition (Herbei & Wegkamp, 2006), uncertainty estimation, detecting covariate shift, active learning, model-based reinforcement learning, and transfer learning. Accordingly, these applications have lead to widespread interest in deep generative models, which take many forms such as variational auto-encoders (VAEs) (Kingma & Welling, 2014; Rezende et al., 2014), generative adversarial networks (GANs) (Goodfel-

---

[*]Corresponding authors: e.nalisnick@eng.cam.ac.uk and balajiln@google.com.
[†]Work done during an internship at DeepMind.

low et al., 2014), auto-regressive models (van den Oord et al., 2016b;a), and invertible latent variable models (Tabak & Turner, 2013). The last two classes—auto-regressive and invertible models—are especially attractive since they offer exact computation of the marginal likelihood, requiring no approximate inference techniques.

In this paper, we investigate if modern deep generative models can be used for anomaly detection, as suggested by Bishop (1994) and the AABI pannel (Blei et al., 2017), expecting a well-calibrated model to assign higher density to the training data than to some other data set. However, we find this to not be the case: when trained on CIFAR-10 (Krizhevsky & Hinton, 2009), VAEs, autoregressive models, and flow-based generative models all assign a higher density to SVHN (Netzer et al., 2011) than to the training data. We find this observation to be quite problematic and unintuitive since SVHN's digit images are so visually distinct from the dogs, horses, trucks, boats, etc. found in CIFAR-10. Yet this phenomenon is not restricted to CIFAR-10 vs SVHN, and we report similar findings for models trained on CelebA and ImageNet. We go on to study these curious observations in flow-based models in particular since they allow for exact marginal density calculations. When the flow is restricted to have constant volume across inputs, we show that the out-of-distribution behavior can be explained in terms of the data's variance and the model's curvature.

To the best of our knowledge, we are the first to report these unintuitive findings for a variety of deep generative models and image data sets. Moreover, our experiments with flow-based models isolate some crucial experimental variables such as the effect of constant-volume vs non-volume-preserving transformations. Lastly, our analysis provides some simple but general expressions for quantifying the gap in the model density between two data sets. We close the paper by urging more study of the out-of-training-distribution properties of deep generative models. Understanding their behaviour in this setting is crucial for their deployment to the real world.

## 2 BACKGROUND

We begin by establishing notation and reviewing the necessary background material. We denote matrices with upper-case and bold letters (e.g. $\boldsymbol{X}$), vectors with lower-case and bold (e.g. $\boldsymbol{x}$), and scalars with lower-case and no bolding (e.g. $x$). As our focus is on generative models, let the collection of all observations be denoted by $\boldsymbol{X} = \{\boldsymbol{x}_n\}_{n=1}^N$ with $\boldsymbol{x}$ representing a vector containing all features and, if present, labels. All $N$ examples are assumed independently and identically drawn from some population $\boldsymbol{x} \sim p^*(\boldsymbol{x})$ (which is unknown) with support denoted $\mathcal{X}$. We define the model density function to be $p(\boldsymbol{x}; \boldsymbol{\theta})$ where $\boldsymbol{\theta} \in \boldsymbol{\Theta}$ are the model parameters, and let the model likelihood be denoted $p(\boldsymbol{X}; \boldsymbol{\theta}) = \prod_{n=1}^N p(\boldsymbol{x}_n; \boldsymbol{\theta})$.

### 2.1 TRAINING NEURAL GENERATIVE MODELS

Given (training) data $\boldsymbol{X}$ and a model class $\{p(\cdot; \boldsymbol{\theta}) : \boldsymbol{\theta} \in \boldsymbol{\Theta}\}$, we are interested in finding the parameters $\boldsymbol{\theta}$ that make the model closest to the true but unknown data distribution $p^*(\boldsymbol{x})$. We can quantify this gap in terms of a Kullback–Leibler divergence (KLD):

$$\mathrm{KLD}[p^*(\boldsymbol{x})||p(\boldsymbol{x}; \boldsymbol{\theta})] = \int p^*(\boldsymbol{x}) \log \frac{p^*(\boldsymbol{x})}{p(\boldsymbol{x}; \boldsymbol{\theta})} \, d\boldsymbol{x} \approx -\frac{1}{N} \log p(\boldsymbol{X}; \boldsymbol{\theta}) - \mathbb{H}[p^*] \qquad (1)$$

where the first term in the right-most expression is the average log-likelihood and the second is the entropy of the true distribution. As the latter is a fixed constant, minimizing the KLD amounts to finding the parameter settings that maximize the data's log density: $\boldsymbol{\theta}^* = \arg\max_{\boldsymbol{\theta}} \log p(\boldsymbol{X}; \boldsymbol{\theta}) = \arg\max_{\boldsymbol{\theta}} \sum_{n=1}^N \log p(\boldsymbol{x}_n; \boldsymbol{\theta})$. Note that $p(\boldsymbol{x}_n; \boldsymbol{\theta})$ alone does not have any interpretation as a probability. To extract probabilities from the model density, we need to integrate over some region $\boldsymbol{\Omega}$: $P(\boldsymbol{\Omega}) = \int_{\boldsymbol{\Omega}} p(\boldsymbol{x}; \boldsymbol{\theta}) d\boldsymbol{x}$. Adding noise to the data during model optimization can mock this integration step, encouraging the density model to output something nearer to probabilities (Theis et al., 2016):

$$\log \int p(\boldsymbol{x}_n + \boldsymbol{\delta}; \boldsymbol{\theta}) p(\boldsymbol{\delta}) \, d\boldsymbol{\delta} \geq \mathbb{E}_{\boldsymbol{\delta}} \left[ \log p(\boldsymbol{x}_n + \boldsymbol{\delta}; \boldsymbol{\theta}) \right] \approx \log p(\boldsymbol{x}_n + \tilde{\boldsymbol{\delta}}; \boldsymbol{\theta})$$

where $\tilde{\boldsymbol{\delta}}$ is a sample from $p(\boldsymbol{\delta})$. The resulting objective is a lower-bound, making it a suitable optimization target. All models in all of the experiments that we report are trained with input noise.

Due to this ambiguity between densities and probabilities, we call the quantity $\log p(\boldsymbol{X} + \tilde{\boldsymbol{\Delta}}; \boldsymbol{\theta})$ a 'log-likelihood,' even if $\boldsymbol{X}$ is drawn from a distribution unlike the training data.

Regarding the choice of density model, we could choose one of the standard density functions for $p(\boldsymbol{x}_n; \boldsymbol{\theta})$, e.g. a Gaussian, but these may not be suitable for modeling the complex, high-dimensional data sets we often observe in the real world. Hence, we want to parametrize the model density with some high-capacity function $f$, which is usually chosen to be a neural network. That way the model has a somewhat compact representation and can be optimized via gradient ascent. We experiment with three variants of neural generative models: autoregressive, latent variable, and invertible. In the first class, we study the *PixelCNN* (van den Oord et al., 2016b), and due to space constraints, we refer the reader to van den Oord et al. (2016b) for its definition. As a representative of the second class, we use a *VAE* (Kingma & Welling, 2014; Rezende et al., 2014). See Rosca et al. (2018) for descriptions of the precise versions we use. Lastly, invertible flow-based generative models are the third class. We define them in detail below since we study them with the most depth.

## 2.2   GENERATIVE MODELS VIA CHANGE OF VARIABLES

The VAE and many other generative models are defined as a joint distribution between the observed and latent variables. However, another path forward is to perform a *change of variables*. In this case $\boldsymbol{x}$ and $\boldsymbol{z}$ are one and the same, and there is no longer any notion of a product space $\mathcal{X} \times \mathcal{Z}$. Let $f : \mathcal{X} \mapsto \mathcal{Z}$ be a diffeomorphism from the data space $\mathcal{X}$ to a latent space $\mathcal{Z}$. Using $f$ then allows us to compute integrals over $\boldsymbol{z}$ as an integral over $\boldsymbol{x}$ and vice versa:

$$\int_{\boldsymbol{z}} p_z(\boldsymbol{z}) \, d\boldsymbol{z} = \int_{\boldsymbol{x}} p_z(f(\boldsymbol{x})) \left| \frac{\partial \boldsymbol{f}}{\partial \boldsymbol{x}} \right| \, d\boldsymbol{x} = \int_{\boldsymbol{x}} p_x(\boldsymbol{x}) \, d\boldsymbol{x} = \int_{\boldsymbol{z}} p_x(f^{-1}(\boldsymbol{z})) \left| \frac{\partial \boldsymbol{f}^{-1}}{\partial \boldsymbol{z}} \right| \, d\boldsymbol{z} \quad (2)$$

where $|\partial \boldsymbol{f} / \partial \boldsymbol{x}|$ and $|\partial \boldsymbol{f}^{-1} / \partial \boldsymbol{z}|$ are known as the *volume elements* as they adjust for the volume change under the alternate measure. Specifically, when the change is w.r.t. coordinates, the volume element is the determinant of the diffeomorphism's Jacobian matrix, which we denote as $|\partial \boldsymbol{f} / \partial \boldsymbol{x}|$.

The change of variables formula is a powerful tool for generative modeling as it allows us to define a distribution $p(\boldsymbol{x})$ entirely in terms of an auxiliary distribution $p(\boldsymbol{z})$, which we are free to choose, and $f$. Denote the parameters of the change of variables model as $\boldsymbol{\theta} = \{\boldsymbol{\phi}, \boldsymbol{\psi}\}$ with $\boldsymbol{\phi}$ being the diffeomorphism's parameters, i.e. $f(\boldsymbol{x}; \boldsymbol{\phi})$, and $\boldsymbol{\psi}$ being the auxiliary distribution's parameters, i.e. $p(\boldsymbol{z}; \boldsymbol{\psi})$. We can perform maximum likelihood estimation for the model as follows:

$$\boldsymbol{\theta}^* = \arg\max_{\boldsymbol{\theta}} \log p_x(\boldsymbol{X}; \boldsymbol{\theta}) = \arg\max_{\boldsymbol{\phi}, \boldsymbol{\psi}} \sum_{n=1}^{N} \log p_z(f(\boldsymbol{x}_n; \boldsymbol{\phi}); \boldsymbol{\psi}) + \log \left| \frac{\partial \boldsymbol{f}_{\boldsymbol{\phi}}}{\partial \boldsymbol{x}_n} \right|. \quad (3)$$

Optimizing $\boldsymbol{\psi}$ must be done carefully so as to not result in a trivial model. For instance, optimization could make $p(\boldsymbol{z}; \boldsymbol{\psi})$ close to uniform if there are no constraints on its variance. For this reason, most implementations leave $\boldsymbol{\psi}$ as fixed (usually a standard Gaussian) in practice. Likewise, we assume it as fixed from here forward, thus omitting $\boldsymbol{\psi}$ from equations to reduce notational clutter. After training, samples can be drawn from the model via the inverse transform: $\tilde{\boldsymbol{x}} = f^{-1}(\tilde{\boldsymbol{z}}; \boldsymbol{\phi}), \quad \tilde{\boldsymbol{z}} \sim p(\mathbf{z})$.

For the particular form of $f$, most work to date has constructed the bijection from *affine coupling layers* (ACLs) (Dinh et al., 2017), which transform $\boldsymbol{x}$ by way of translation and scaling operations. Specifically, ACLs take the form: $f_{\text{ACL}}(\boldsymbol{x}; \boldsymbol{\phi}) = [\exp\{s(\boldsymbol{x}_{d:}; \boldsymbol{\phi}_s)\} \odot \boldsymbol{x}_{:d} + t(\boldsymbol{x}_{d:}; \boldsymbol{\phi}_t), \boldsymbol{x}_{d:}]$, where $\odot$ denotes an element-wise product. This transformation, firstly, splits the input vector in half, i.e. $\boldsymbol{x} = [\boldsymbol{x}_{:d}, \boldsymbol{x}_{d:}]$ (using Python list syntax). Then the second half of the vector is fed into two arbitrary neural networks (possibly with tied parameters) whose outputs are denoted $t(\boldsymbol{x}_{d:}; \boldsymbol{\phi}_t)$ and $s(\boldsymbol{x}_{d:}; \boldsymbol{\phi}_s)$, with $\boldsymbol{\phi}$. being the collection of weights and biases. Finally, the output is formed by (1) *scaling* the first half of the input by one neural network output, i.e. $\exp\{s(\boldsymbol{x}_{d:}; \boldsymbol{\phi}_s)\} \odot \boldsymbol{x}_{:d}$, (2) *translating* the result of the scaling operation by the second neural network output, i.e. $(\cdot) + t(\boldsymbol{x}_{d:}; \boldsymbol{\phi}_t)$, and (3) *copying* the second half of $\boldsymbol{x}$ forward, making it the second half of $f_{\text{ACL}}(\boldsymbol{x}; \boldsymbol{\phi})$, i.e. $f_{d:} = \boldsymbol{x}_{d:}$. ACLs are stacked to make rich hierarchical transforms, and the latent representation $\boldsymbol{z}$ is output from this composition, i.e. $\boldsymbol{z}_n = f(\boldsymbol{x}_n; \boldsymbol{\phi})$. A permutation operation is required between ACLs to ensure the same elements are not repeatedly used in the copy operations. We use $f$ without subscript to denote the complete transform and overload the use of $\boldsymbol{\phi}$ to denote the parameters of all constituent layers.

This class of transform is known as *non-volume preserving* (NVP) (Dinh et al., 2017) since the volume element does not necessarily evaluate to one and can vary with each input $\boldsymbol{x}$. Although

non-zero, the log determinant of the Jacobian is still tractable: $\log |\partial \boldsymbol{f}_\phi / \partial \boldsymbol{x}| = \sum_{j=d}^{D} s_j(\boldsymbol{x}_{d:}; \boldsymbol{\phi}_s)$. A diffeomorphic transform can also be defined with just translation operations, as was done in earlier work by Dinh et al. (2015), and this transformation is *volume preserving* (VP) since the volume term is one and thus has no influence in the likelihood calculation. We will examine another class of flows we term *constant-volume* (CV) since the volume, while not preserved, is constant across all $\boldsymbol{x}$. Appendix A provides additional details on implementing flow-based generative models.

## 3  MOTIVATING OBSERVATIONS

Given the impressive advances of deep generative models, we sought to test their ability to quantify when an input comes from a different distribution than that of the training set. This calibration w.r.t. out-of-distribution data is essential for applications such as safety—if we were using the generative model to filter the inputs to a discriminative model—and for active learning. For the experiment, we trained the same Glow architecture described in Kingma & Dhariwal (2018)—except small enough that it could fit on one GPU[1]—on FashionMNIST and CIFAR-10. Appendix A provides additional implementation details. We then calculated the *log-likelihood* (higher value is better) and *bits-per-dimension* (BPD, lower value is better)[2] of the test split of two different data sets of the same dimensionality—MNIST ($28 \times 28$) and SVHN ($32 \times 32 \times 3$) respectively. We expect the models to assign a lower probability to this data because they were not trained on it. Samples from the Glow models trained on each data set are shown in Figure 13 in the Appendix.

| Data Set | Avg. Bits Per Dimension | Data Set | Avg. Bits Per Dimension |
|---|---|---|---|
| *Glow Trained on FashionMNIST* | | *Glow Trained on CIFAR-10* | |
| FashionMNIST-Train | 2.902 | CIFAR10-Train | 3.386 |
| FashionMNIST-Test | 2.958 | CIFAR10-Test | 3.464 |
| MNIST-Test | **1.833** | SVHN-Test | **2.389** |
| *Glow Trained on MNIST* | | *Glow Trained on SVHN* | |
| MNIST-Test | 1.262 | SVHN-Test | 2.057 |

Figure 1: *Testing Out-of-Distribution.* Log-likelihood (expressed in bits per dimension) calculated from Glow (Kingma & Dhariwal, 2018) on MNIST, FashionMNIST, SVHN, CIFAR-10.

Beginning with FashionMNIST vs MNIST, the left subtable of Figure 1 shows the average BPD of the training data (FashionMNIST-Train), the in-distribution test data (FashionMNIST-Test), and the out-of-distribution data (MNIST-Test). We see a peculiar result: the MNIST split has the *lowest* BPD, more than one bit less than the FashionMNIST train and test sets. To check if this is due to outliers skewing the average, we report a (normalized) histogram in Figure 2 (a) of the log-likelihoods for the three splits. We see that MNIST (red bars) is clearly and systematically shifted to the RHS of the plot (highest likelihood).

Moving on to CIFAR-10 vs SVHN, the right subtable of Figure 1 again reports the BPD of the training data (CIFAR10-Train), the in-distribution test data (CIFAR10-Test), and the out-of-distribution data (SVHN-Test). We again see the phenomenon: the SVHN BPD is one bit *lower* than that of both in-distribution data sets. Figure 2 (b) shows a similar histogram of the log-likelihoods. Clearly the SVHN examples (red bars) have a systematically higher likelihood, and therefore the result is not caused by any outliers.

Subfigures (c) and (d) of Figure 2 show additional results for CelebA and ImageNet. When trained on CelebA, Glow assigns a higher likelihood to SVHN (red bars), a data set the model has never seen before. Similarly, when trained on ImageNet, Glow assigns a higher likelihood to the test splits of SVHN (red), CIFAR-10 (yellow), and CIFAR-100 (green). The difference is quite drastic in the case of SVHN (red) but modest for the two CIFAR splits. This phenomenon is not symmetric. CIFAR-10 does not have a higher likelihood under a Glow trained on SVHN; see Figure 6 in Appendix B

---

[1] Although we use a smaller model, it still produces good samples, which can be seen in Figure 13 of the Appendix, and competitive BPD (CIFAR-10: 3.46 for ours vs 3.35 for theirs).

[2] See Theis et al. (2016, Section 3.1) for the definitions of log-likelihood and bits-per-dimension.

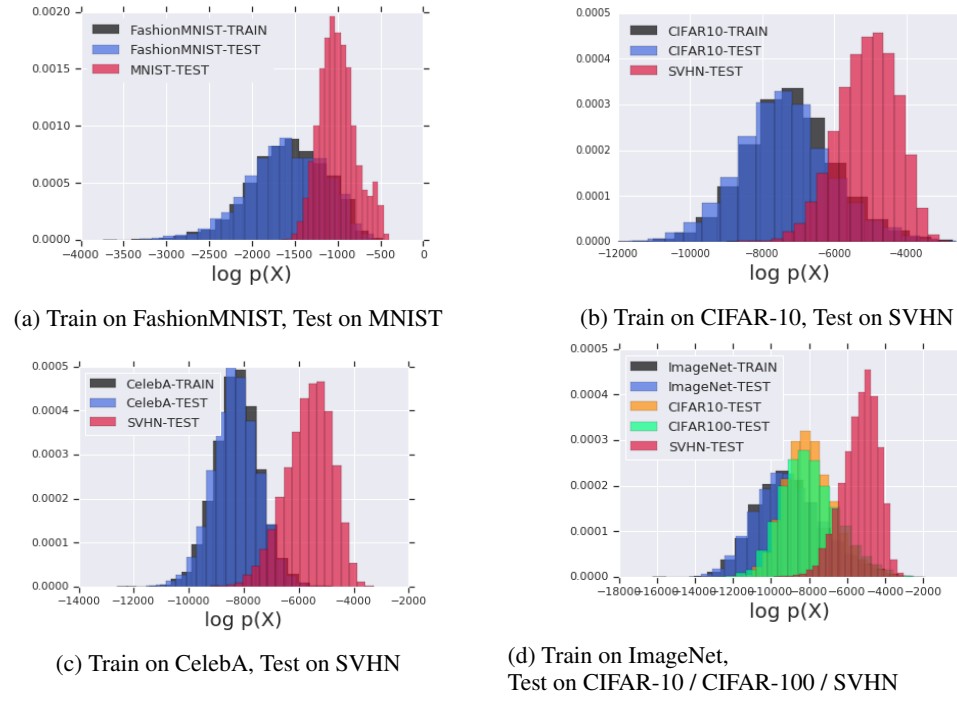

(a) Train on FashionMNIST, Test on MNIST

(b) Train on CIFAR-10, Test on SVHN

(c) Train on CelebA, Test on SVHN

(d) Train on ImageNet,
Test on CIFAR-10 / CIFAR-100 / SVHN

Figure 2: Histogram of Glow log-likelihoods for FashionMNIST vs MNIST (a), CIFAR-10 vs SVHN (b), CelebA vs SVHN (c), and ImageNet vs CIFAR-10 / CIFAR-100 / SVHN (d).

for these results. We report results only for Glow, but we observed the same behavior for RNVP transforms (Dinh et al., 2017).

We next tested if the phenomenon occurs for other common deep generative models: PixelCNNs and VAEs. We do not include GANs in the comparison since evaluating their likelihood is an open problem. Figure 3 reports the same histograms as above for these models, showing the distribution of $\log p(\boldsymbol{x})$ evaluations for FashionMNIST vs MNIST (a, b) and CIFAR-10 vs SVHN (c, d). The training splits are again denoted with black bars, and the test splits with blue, and the out-of-distribution splits with red. The red bars are shifted to the right in all four plots, signifying the behavior exists in spite of the differences between model classes.

## 4   DIGGING DEEPER INTO THE FLOW-BASED MODEL

While we observed the out-of-distribution phenomenon for PixelCNN, VAE, and Glow, now we narrow our investigation to just the class of invertible generative models. The rationale is that they allow for better experimental control as, firstly, they can compute exact marginal likelihoods (unlike VAEs), and secondly, the transforms used in flow-based models have Jacobian constraints that simplify the analysis we present in Section 5. To further analyze the high likelihood of the out-of-distribution (non-training) samples, we next report the contributions to the likelihood of each term in the change-of-variables formula. At first this suggested the volume element was the primary cause of SVHN's high likelihood, but further experiments with constant-volume flows show the problem exists with them as well.

**Decomposing the change-of-variables objective.**   To further examine this curious phenomenon, we inspect the change-of-variables objective itself, investigating if one or both terms give the out-of-distribution data a higher value. We report the constituent $\log p(\boldsymbol{z})$ and $\log |\partial \boldsymbol{f}_\phi / \partial \boldsymbol{x}|$ terms for NVP-Glow in Figure 4, showing histograms for $\log p(\boldsymbol{z})$ in subfigure (a) and for $\log |\partial \boldsymbol{f}_\phi / \partial \boldsymbol{x}|$ in subfigure (b). We see that $p(\boldsymbol{z})$ behaves mostly as expected. The red bars (SVHN) are clearly shifted to the left, representing lower likelihoods under the latent distribution. Moving on to the volume element, this term seems to cause SVHN's higher likelihood. Subfigure (b) shows that all of the

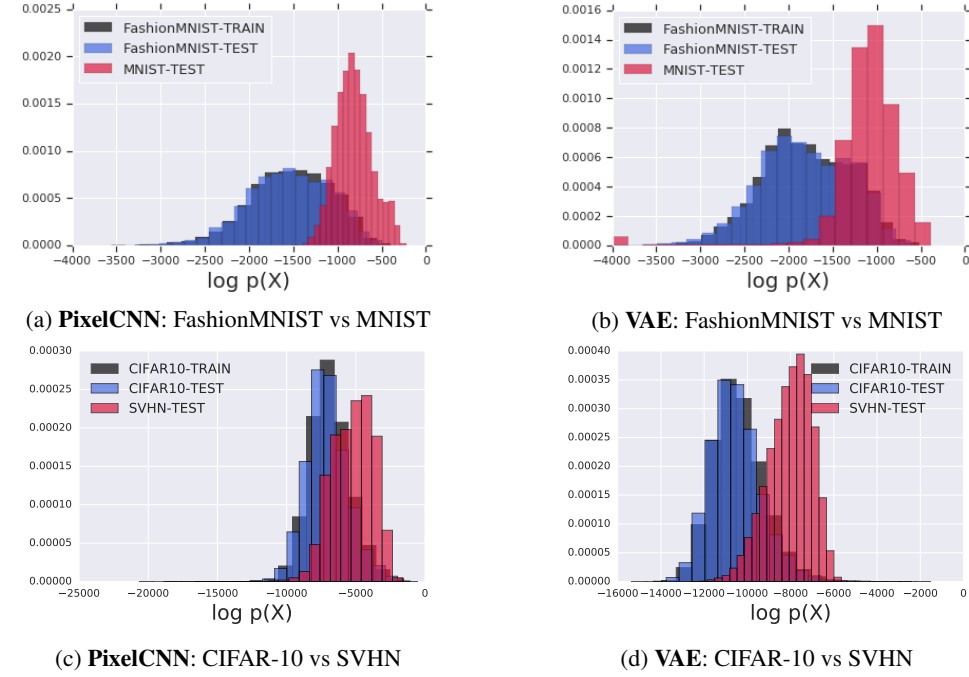

Figure 3: *PixelCNN and VAE*. Log-likelihoods calculated by PixelCNN (a, c) and VAE (b, d) on FashionMNIST vs MNIST (a, b) and CIFAR-10 vs SVHN (c, d). VAE models are the convolutional categorical variant described by Rosca et al. (2018).

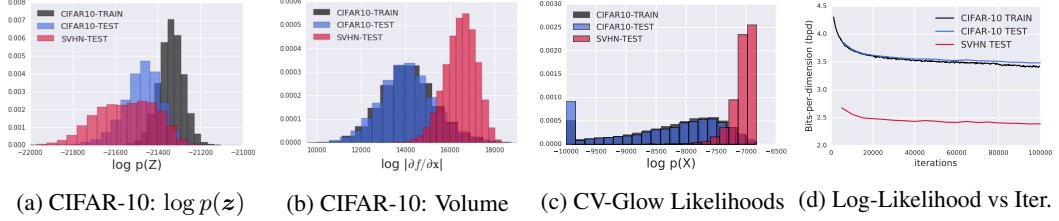

Figure 4: *Decomposing the Likelihood of NVP-Glow / CV-Glow Results.* The histograms in (a) and (b) show NVP-Glow's log-likelihood decomposed into contributions from the $z$-distribution and volume element, respectively, for CIFAR-10 vs SVHN. Subfigure (c) shows log-likelihood evaluations for constant-volume (CV) Glow, again when trained on CIFAR-10 and tested on SVHN. Subfigure (d) reports NVP-Glow's BPD over the course of training, showing that the phenomenon happens throughout and could not be prevented by early stopping.

SVHN log-volume evaluations (red) are conspicuously shifted to the right—to higher values—when compared to CIFAR-10's (blue and black). Since SVHN's $p(z)$ evaluations are only slightly less than CIFAR-10's, the volume term dominates, resulting in SVHN having a higher likelihood.

**Is the volume the culprit?** In addition to the empirical evidence against the volume element, we notice that one of the terms in the change-of-variables objective—by rewarding the maximization of the Jacobian determinant—encourages the model to *increase* its sensitivity to perturbations in $\mathcal{X}$. This behavior starkly contradicts a long history of derivative-based regularization penalties that reward the model for *decreasing* its sensitivity to input directions. For instance, Girosi et al. (1995) and Rifai et al. (2011) propose penalizing the Frobenius norm of a neural network's Jacobian for classifiers and autoencoders respectively. See Appendix C for more analysis of the log volume element.

To experimentally control for the effect of the volume term, we trained Glow with constant-volume (CV) transformations. We modify the affine layers to use only translation operations (Dinh et al., 2015) but keep the $1 \times 1$ convolutions as is. The log-determinant-Jacobian is then $HW \sum_k \log |U_k|$,

where $|\boldsymbol{U}_k|$ is the determinant of the convolutional weights $\boldsymbol{U}_k$ for the $k$th flow. This makes the volume element constant across all inputs $\boldsymbol{x}$, allowing us to isolate its effect while still keeping the model expressive. Subfigures (c) and (d) of Figure 4 show the results for this model, which we term *CV-Glow* (constant-volume Glow). Subfigure (c) shows a histogram of the $\log p(\boldsymbol{x})$ evaluations, just as shown before in Figure 2, and we see that SVHN (red) still achieves a higher likelihood (lower BPD) than the CIFAR-10 training set. Subfigure (d) shows the SVHN vs CIFAR-10 BPD over the course of training for NVP-Glow. Notice that there is no cross-over point in the curves.

**Other experiments: random and constant images, ensembles.** Other work on generative models (Sønderby et al., 2017; van den Oord et al., 2018) has noted that they often assign the highest likelihood to constant inputs. We also test this case, reporting the BPD in Appendix Figure 8 for NVP-Glow models. We find constant inputs have the highest likelihood for our models as well: 0.589 BPD for CIFAR-10. We also include in the table the BPD of random inputs for comparison.

We also hypothesized that averaging over the parameters may mitigate the phenomenon. While integration over the entire parameter space would be ideal, this is analytically and computationally difficult for Glow. Lakshminarayanan et al. (2017) show that deep ensembles can guard against over-confidence for anomalous inputs while being more practical to implement. We opted for this approach, training five Glow models independently and averaging their likelihoods to evaluate test data. Each model was given a different initialization of the parameters to help diversify the ensemble. Figure 9 in Appendix F reports a histogram of the $\log p(\boldsymbol{x})$ evaluations when averaging over the ensemble. We see nearly identical results: SVHN is still assigned a higher likelihood than the CIFAR-10 training data.

## 5    SECOND ORDER ANALYSIS

In this section, we aim to provide a more direct analysis of when another distribution might have higher likelihood than the one used for training. We propose analyzing the phenomenon by way of linearizing the difference in expected log-likelihoods. This approach undoubtedly gives a crude approximation, but as we show below, it agrees with and gives insight into some of the observations reported above. Consider two distributions: the training distribution $\boldsymbol{x} \sim p^*$ and some dissimilar distribution $\boldsymbol{x} \sim q$ also with support on $\mathcal{X}$. For a given generative model $p(\boldsymbol{x}; \boldsymbol{\theta})$, the adversarial distribution $q$ will have a higher likelihood than the training data's if $\mathbb{E}_q[\log p(\boldsymbol{x}; \boldsymbol{\theta})] - \mathbb{E}_{p^*}[\log p(\boldsymbol{x}; \boldsymbol{\theta})] > 0$. This expression is hard to analyze directly so we perform a second-order expansion of the log-likelihood around an interior point $\boldsymbol{x}_0$. Applying the expansion $\log p(\boldsymbol{x}; \boldsymbol{\theta}) \approx \log p(\boldsymbol{x}_0; \boldsymbol{\theta}) + \nabla_{\boldsymbol{x}_0} \log p(\boldsymbol{x}_0; \boldsymbol{\theta})^T (\boldsymbol{x} - \boldsymbol{x}_0) + \frac{1}{2} \text{Tr}\{\nabla^2_{\boldsymbol{x}_0} \log p(\boldsymbol{x}_0; \boldsymbol{\theta})(\boldsymbol{x} - \boldsymbol{x}_0)(\boldsymbol{x} - \boldsymbol{x}_0)^T\}$ to both likelihoods, taking expectations, and canceling the common terms, we have:

$$0 < \mathbb{E}_q[\log p(\boldsymbol{x}; \boldsymbol{\theta})] - \mathbb{E}_{p^*}[\log p(\boldsymbol{x}; \boldsymbol{\theta})]$$
$$\approx \nabla_{\boldsymbol{x}_0} \log p(x_0; \boldsymbol{\theta})^T (\mathbb{E}_q[\boldsymbol{x}] - \mathbb{E}_{p^*}[\boldsymbol{x}]) + \frac{1}{2} \text{Tr}\{\nabla^2_{\boldsymbol{x}_0} \log p(\boldsymbol{x}_0; \boldsymbol{\theta})(\boldsymbol{\Sigma}_q - \boldsymbol{\Sigma}_{p^*})\} \tag{4}$$

where $\boldsymbol{\Sigma} = \mathbb{E}\left[(\boldsymbol{x} - \boldsymbol{x}_0)(\boldsymbol{x} - \boldsymbol{x}_0)^T\right]$, the covariance matrix, and $\text{Tr}\{\cdot\}$ is the trace operation. Since the expansion is accurate only locally around $\boldsymbol{x}_0$, we next assume that $\mathbb{E}_q[\boldsymbol{x}] = \mathbb{E}_{p^*}[\boldsymbol{x}] = \boldsymbol{x}_0$. While this at first glance may seem like a strong assumption, it is not too removed from practice since data is usually centered before being fed to the model. For SVHN and CIFAR-10 in particular, we find this assumption to hold; see Figure 5 (a) for the empirical means of each dimension of CIFAR-10 (green) and SVHN (orange). All of SVHN's means fall within the empirical range of CIFAR-10's, and the maximum difference between any dimension is less than 38 pixel values. Assuming equal means, we then have:

$$0 < \mathbb{E}_q[\log p(\boldsymbol{x}; \boldsymbol{\theta})] - \mathbb{E}_{p^*}[\log p(\boldsymbol{x}; \boldsymbol{\theta})] \approx \frac{1}{2} \text{Tr}\{\nabla^2_{\boldsymbol{x}_0} \log p(\boldsymbol{x}_0; \boldsymbol{\theta})(\boldsymbol{\Sigma}_q - \boldsymbol{\Sigma}_{p^*})\}$$
$$= \frac{1}{2} \text{Tr}\left\{\left[\nabla^2_{\boldsymbol{x}_0} \log p_z(f(\boldsymbol{x}_0; \boldsymbol{\phi})) + \nabla^2_{\boldsymbol{x}_0} \log \left|\frac{\partial \boldsymbol{f}_\phi}{\partial \boldsymbol{x}_0}\right|\right](\boldsymbol{\Sigma}_q - \boldsymbol{\Sigma}_{p^*})\right\}, \tag{5}$$

where the second line assumes the generative model to be flow-based.

**Analysis of CV-Glow.** We use the expression in Equation 5 to analyze the behavior of CV-Glow on CIFAR-10 vs SVHN, seeing if the difference in likelihoods can be explained by the model curvature

and data's second moment. The second derivative terms simplify considerably for CV-Glow with a spherical latent density. Given a $C \times C$ kernel $\boldsymbol{U}_k$, with $k$ indexing the flow and $C$ the number of input channels, the derivatives are $\partial f_{h,w,c}/\partial x_{h,w,c} = \prod_k \sum_{j=1}^{C} u_{k,c,j}$, with $h$ and $w$ indexing the spatial height and width and $j$ the columns of the $k$th flow's $1 \times 1$ convolutional kernel. The second derivative is then $\partial^2 f_{h,w,c}/\partial x_{h,w,c}^2 = 0$, which allows us to write

$$\text{Tr}\left\{ \left[ \nabla_{\boldsymbol{x}_0}^2 \log p(\boldsymbol{x}_0; \boldsymbol{\theta}) \right] (\boldsymbol{\Sigma}_q - \boldsymbol{\Sigma}_{p^*}) \right\}$$

$$= \frac{\partial^2}{\partial z^2} \log p(\boldsymbol{z}; \boldsymbol{\psi}) \sum_{c=1}^{C} \left( \prod_{k=1}^{K} \sum_{j=1}^{C} u_{k,c,j} \right)^2 \sum_{h,w} (\sigma_{q,h,w,c}^2 - \sigma_{p^*,h,w,c}^2).$$

The derivation is given in Appendix G. Plugging in the second derivative of the Gaussian's log density—a common choice for the latent distribution in flow models (Dinh et al., 2017; Kingma & Dhariwal, 2018)—and the empirical variances, we have:

$$\mathbb{E}_{\text{SVHN}}[\log p(\boldsymbol{x}; \boldsymbol{\theta})] - \mathbb{E}_{\text{CIFAR-10}}[\log p(\boldsymbol{x}; \boldsymbol{\theta})]$$

$$\approx \frac{-1}{2\sigma_{\boldsymbol{\psi}}^2} \left[ \alpha_1^2(49.6 - 61.9) + \alpha_2^2(52.7 - 59.2) + \alpha_3^2(53.6 - 68.1) \right]$$

$$\tag{6}$$

$$= \frac{1}{2\sigma_{\boldsymbol{\psi}}^2} \left[ \alpha_1^2 \cdot 12.3 + \alpha_2^2 \cdot 6.5 + \alpha_3^2 \cdot 14.5 \right] \geq 0 \quad \text{where} \quad \alpha_c = \prod_{k=1}^{K} \sum_{j=1}^{C} u_{k,c,j}$$

and where $\sigma_{\boldsymbol{\psi}}^2$ is the variance of the latent distribution. We know the final expression is greater than or equal to zero since all $\alpha_c^2 \geq 0$. Equality is achieved only for $\sigma_{\boldsymbol{\psi}}^2 \to \infty$ or in the unusual case of at least one all-zero row in any convolutional kernel for all channels. Thus, the second-order expression does indeed predict we should see a higher likelihood for SVHN than for CIFAR-10. Moreover, we leave the CV-Glow's parameters as constants to emphasize the expression is non-negative *for any parameter setting*. This finding is supported by our observations that using an ensemble of Glows resulted in an almost identical likelihood gap (Figure 9) and that the gap remained relatively constant over the course of training (Figure 4d). Furthermore, the $\partial^2 \log p(\boldsymbol{z}; \boldsymbol{\psi})/\partial z^2$ term would be negative for any log-concave density function, meaning that changing the latent density to Laplace or logistic would not change the result.

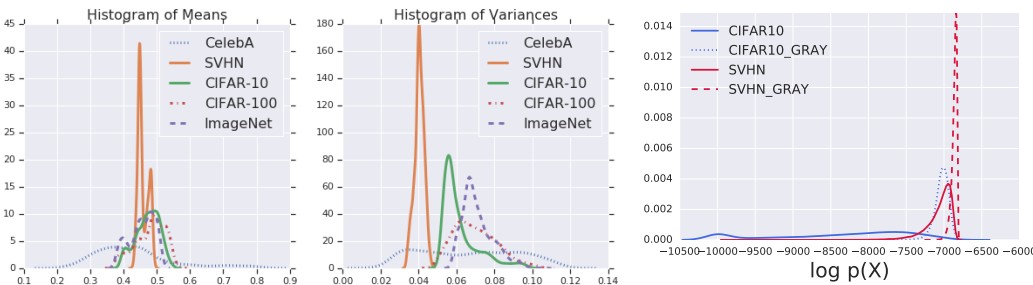

(a) Histogram of per-dimension means and variances (empirical).     (b) Graying images increases likelihood.

Figure 5: *Empirical Distributions and Graying Effect.* Note that pixels are converted from 0-255 scale to 0-1 scale by diving by 256. See Figure 10 for results on datasets of $28 \times 28 \times 1$ images.

Our conclusion is that SVHN simply "sits inside of" CIFAR-10—roughly same mean, smaller variance—resulting in its higher likelihood. This insight also holds true for the additional results presented in subfigures (c) and (d) of Figure 2. Examining Figure 5 (a) again, we see that ImageNet, the CIFARs, and SVHN all have nearly overlapping means and that ImageNet has the highest variance. Therefore we expect SVHN and the CIFARs to have a higher likelihood than ImageNet on an ImageNet-trained model, which is exactly what we observe in Figure 2 (d). Moreover, the degree of the differences in likelihoods agrees with the differences in variances. SVHN clearly has the smallest variance and the largest likelihood. In turn, we can artificially increase the likelihood of a data set by shrinking its variance. For RGB images, shrinking the variance is equivalent to 'graying' the images, i.e. making the pixel values closer to 128. We show in Figure 5 (b) that doing

exactly this improves the likelihood of both CIFAR-10 and SVHN. Reducing the variance of the latent representations has the same effect, which is shown by Figure 12 in the Appendix.

## 6 RELATED WORK

This paper is inspired by and most related to recent work on evaluation of generative models. Worthy of foremost mention is the work of Theis et al. (2016), which showed that high likelihood is neither sufficient nor necessary for the model to produce visually satisfying samples. However, their paper does not consider out-of-distribution inputs. In this regard, there has been much work on *adversarial inputs* (Szegedy et al., 2014). While the term is used broadly, it commonly refers to inputs that have been imperceptibly modified so that the model can no longer provide an accurate output (a mis-classification, usually). Adversarial attacks on generative models have been studied by (at least) Tabacof et al. (2016) and Kos et al. (2018), but these methods of attack require access to the model. We, on the other hand, are interested in model calibration for any out-of-distribution set and especially for common data sets not constructed with any nefarious intentions nor for attack on a particular model. Various papers (Hendrycks & Gimpel, 2017; Lakshminarayanan et al., 2017; Liang et al., 2018) have reported that discriminative neural networks can produce overconfident predictions on out-of-distribution inputs. In a related finding, Lee et al. (2018) reported that it was much harder to recognize an input as out-of-distribution when the classifier was trained on CIFAR-10 in comparison to training on SVHN.

Testing the robustness of deep generative models to out-of-distribution inputs had not been investigated previously, to the best of our knowledge. However, there is work concurrent with ours that has tested their ability to detect anomalous inputs. Shafaei et al. (2018) and Hendrycks et al. (2019) also observe that PixelCNN++ cannot provide reliable outlier detection. Hendrycks et al. (2019) mitigate the CIFAR-10 vs SVHN issue by exposing the model to outliers during training. They do not consider flow-based models. Škvára et al. (2018) experimentally compare VAEs and GANs against k-nearest neighbors (kNNs), showing that VAEs and GANs outperform kNNs only when known outliers can be used for hyperparameter selection. In the work most similar to ours, Choi & Jang (2018) report the same CIFAR-10 vs SVHN phenomenon for Glow—independently confirming our motivating observation. As a fix, they propose training an ensemble of generative models with an adversarial objective and testing for out-of-training-distribution inputs by computing the *Watanabe-Akaike information criterion* via the ensemble. This work is complementary to ours since they focus on providing a detection method whereas we are interested in understanding how and when the phenomenon can arise. The results we present in Equation 6 do not apply to Choi & Jang (2018)'s models since they use scaling operations in their affine coupling layers, making them NVP.

## 7 DISCUSSION

We have shown that comparing the likelihoods of deep generative models alone cannot identify the training set or inputs like it. Therefore we urge caution when using these models with out-of-training-distribution inputs or in unprotected user-facing systems. Moreover, our analysis in Section 5 shows that the CIFAR-10 vs SVHN phenomenon would persist for any constant-volume Glow no matter the parameter values nor the choice of latent density (as long as it is log-concave). While we cannot conclude that this is a pathology in deep generative models, it does suggest the need for further work on generative models and their evaluation. The models we tested seem to be capturing low-level statistics rather than high-level semantics, and better inductive biases, optimization procedures, or uncertainty quantification may be necessary. Yet, deep generative models can detect out-of-distribution inputs when using alternative metrics (Choi & Jang, 2018) and modified training procedures (Hendrycks et al., 2019). The problem then may be a fundamental limitation of high-dimensional likelihoods. Until these open problems are better understood, we must temper the enthusiasm with which we preach the benefits of deep generative models.

### ACKNOWLEDGMENTS

We thank Aaron van den Oord, Danilo Rezende, Eric Jang, Florian Stimberg, Josh Dillon, Mihaela Rosca, Rui Shu, Sander Dieleman, and the anonymous reviewers for their helpful feedback and discussions.

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

# A  ADDITIONAL IMPLEMENTATION DETAILS

## A.1  FLOW-BASED MODELS

We have described the core building blocks of invertible generative models above, but there are several other architectural choices required in practice. Due to space requirements, we only describe them briefly, referring the reader to the original papers for details. In the most recent extension of this line of work, Kingma & Dhariwal (2018) propose the *Glow* architecture, with its foremost contribution being the use of $1 \times 1$ convolutions in place of discrete permutation operations. Convolutions of this form can be thought of as a relaxed but generalized permutation, having all the representational power of the discrete version with the added benefit of parameters amenable to gradient-based training. As the transformation function becomes deeper, it becomes prone to the same scale pathologies as deep neural networks and therefore requires a normalization step of some form. Dinh et al. (2017) propose incorporating batch normalization and describe how to compute its contribution to the log-determinant-Jacobian term. Kingma & Dhariwal (2018) apply a similar normalization, which they call *actnorm*, but it uses trainable parameters instead of batch statistics. Lastly, both Dinh et al. (2017) and Kingma & Dhariwal (2018) use *multi-scale* architectures that factor out variables at regular intervals, copying them forward to the final latent representation. This gradually reduces the dimensionality of the transformations, improving upon computational costs.

For our MNIST experiments, we used a *Glow* architecture of 2 blocks of 16 affine coupling layers, squeezing the spatial dimension in between the 2 blocks. For our CIFAR experiments, we used 3 blocks of 8 affine coupling blocks, applying the multi-scale architecture between each block. For all coupling blocks, we used a 3-layer Highway network with 200 hidden units for MNIST and 400 hidden units for CIFAR. The networks we trained were shallower than those in Kingma & Dhariwal (2018). We also found that initializing the last layer of the coupling networks to 0 was sufficient to prevent scale pathologies, hence we did not use any form of normalization (batchnorm nor actnorm) for ease of initialization and training in a distributed setting. Convolution kernels were initialized with a truncated normal with variance $1/\sqrt{D}$ where $D$ is fan-in size, except where zero-initialization is prescribed by *Glow*. All networks were trained with the RMSProp optimizer, with a learning rate of $1e - 5$ for 100K steps, decaying by half at 80K and 90K steps. We used a prior with zero mean and unit variance for all experiments. We applied L2 regularization of $5e - 2$ to CIFAR experiments. All experiments used batch size 32.

For CV-Glow, we used additive rather than affine coupling blocks, which removes the influence of coupling blocks on the log-det-Jacobian term. The volume change of the $1 \times 1$ convolutions depend on its weights rather than the input, so the network has a constant volume change.

## A.2  PIXELCNN

We trained a GatedPixelCNN with a categorical distribution. For FashionMNIST, we used a network with 5 gated layers with 32 features, and the final skip connection layer with size 256. We trained for 100K steps with the Adam optimizer and an initial learning rate of $1e - 4$, and decaying at 80K and 90K steps. For CIFAR experiments, we used 15 dated layers with 128 features, and a skip connection size of 1024. Convolutions were initialized with Tensorflow's variance scaling initializer with a uniform distribution and a scale of 1. We trained for 200K steps with the RMSProp optimizer with an initial learning rate of $1e - 4$, decaying by $1/3$ at 120K, 180K, and 195K steps. All experiments used batch size 32.

## A.3  VARIATIONAL AUTO-ENCODERS

We refer to (Rosca et al., 2018, Appendix K) for additional details on the VAEs used in our CIFAR experiments. We used the CIFAR configurations without modification. For FashionMNIST, we used the encoder given in Table 4 of (Rosca et al., 2018, Appendix K, Table 4) and a decoder composed of one linear layer with $7 * 7 * 64$ hidden units, followed by a reshape and three transposed convolutions of feature sizes 32, 32, 256 and strides 2, 2, 1. Weights were initialized with a normal distribution with variance 0.02. We trained for 200K steps using the RMSProp optimizer, with a constant learning rate of $1e - 4$.

## B    RESULTS ILLUSTRATING ASYMMETRIC BEHAVIOR

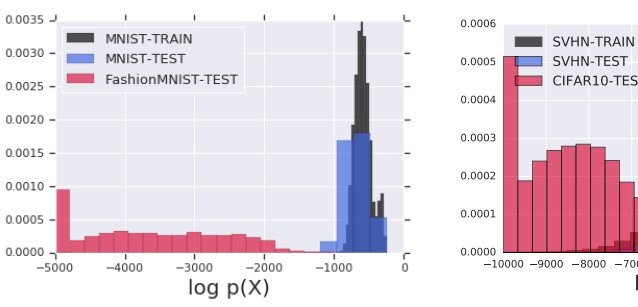

(a) Train on MNIST, Test on FashionMNIST          (b) Train on SVHN, Test on CIFAR-10

Figure 6: Histogram of Glow log-likelihoods for MNIST vs FashionMNIST and SVHN vs CIFAR-10. Note that the model trained on SVHN (MNIST) is able to assign lower likelihood to CIFAR-10 (FashionMNIST), which illustrates the asymmetry compared to Figure 2.

## C    ANALYZING THE CHANGE-OF-VARIABLES FORMULA AS AN OPTIMIZATION FUNCTION

Consider the intuition underlying the volume term in the change of variables objective (Equation 3). As we are maximizing the Jacobian's determinant, it means that the model is being encouraged to maximize the $\partial \boldsymbol{f}_j / \partial x_j$ partial derivatives. In other words, the model is rewarded for making the transformation sensitive to small changes in $\boldsymbol{x}$. This behavior starkly contradicts a long history of derivative-based regularization penalties. Dating back at least to (Girosi et al., 1995), *penalizing the Frobenius norm of a neural network's Jacobian—which upper bounds the volume term[3]—has been shown to improve generalization. This agrees with intuition since we would like the model to be insensitive to small changes in the input, which are likely noise. Moreover, Bishop (1995) showed that training a network under additive Gaussian noise is equivalent to Jacobian regularization, and Rifai et al. (2011) proposed *contractive autoencoders*, which penalize the Jacobian-norm of the encoder. Allowing invertible generative models to maximize the Jacobian term without constraint suggests, at minimum, that these models will not learn robust representations.

**Limiting Behavior.**    We next attempt to quantify the limiting behavior of the log volume element. Let us assume, for the purposes of a general treatment, that the bijection $f_\phi$ is an $L$-Lipschitz function. Both terms in Equation 3 can be bounded as follows:

$$\log p(\boldsymbol{x}; \boldsymbol{\theta}) = \underbrace{\log p_z(f(\boldsymbol{x}; \boldsymbol{\phi}))}_{\mathcal{O}(\max_{\boldsymbol{z}} \log p_z(\boldsymbol{z}))} + \underbrace{\log \left| \frac{\partial \boldsymbol{f}_\phi}{\partial \boldsymbol{x}} \right|}_{\mathcal{O}(D \log L)} \leq \max_{\boldsymbol{z}} \log p_z(\boldsymbol{z}) + D \log L \tag{7}$$

where $L$ is the Lipschitz constant, $D$ the dimensionality, and $\mathcal{O}(\max_{\boldsymbol{z}} \log p_z(\boldsymbol{z}))$ an expression for the (log) mode of $p(\boldsymbol{z})$. We will make this mode term for concrete for Gaussian distributions below. The bound on the volume term follows from Hadamard's inequality:

$$\log \left| \frac{\partial \boldsymbol{f}_\phi}{\partial \boldsymbol{x}} \right| \leq \log \prod_{j=1}^{D} \left| \frac{\partial \boldsymbol{f}_\phi}{\partial \boldsymbol{x}} \boldsymbol{e}_j \right| \leq \log(L \, |\boldsymbol{e}.|)^D = D \log L$$

where $\boldsymbol{e}_j$ is an eigenvector. While this expression is too general to admit any strong conclusions, we can see from it that the 'peakedness' of the distribution represented by the mode must keep pace with the Lipschitz constant, especially as dimensionality increases, in order for both terms to contribute equally to the objective.

We can further illuminate the connection between $L$ and the concentration of the latent distribution through the following proposition:

---

[3]It is easy to show the upper bound via Hadamard's inequality: $\det \partial \boldsymbol{f} / \partial \boldsymbol{x} \leq ||\partial \boldsymbol{f} / \partial \boldsymbol{x}||_F$.

**Proposition 1.** *Assume $x \sim p^*$ is distributed with moments $\mathbb{E}[x] = \mu_x$ and $Var[x] = \sigma_x^2$. Moreover, let $f : \mathcal{X} \mapsto \mathcal{Z}$ be L-Lipschitz and $f(\mu_x) = \mu_z$. We then have the following concentration inequality for some constant $\delta$:*

$$P\left(|f(x) - \mu_z| \geq \delta\right) \leq \frac{L^2 \sigma_x^2}{\delta^2}.$$

*Proof*: From the fact that $f$ is $L$-Lipschitz, we know $|f(x) - \mu_z| \leq L\left|x - f^{-1}(\mu_z)\right|$. Assuming $\mu_x = f^{-1}(\mu_z))$, we can apply Chebyshev's inequality to the RHS: $Pr(L\left|x - f^{-1}(\mu_z)\right| \geq \delta) \leq L^2 \sigma_x^2 / \delta^2$. Since $L\left|x - f^{-1}(\mu_z)\right| \geq |f(x) - \mu_z|$, we can plug the RHS into the inequality and the bound will continue to hold.

From the inequality we can see that the latent distribution can be made more concentrated by decreasing $L$ and/or the data's variance $\sigma_x^2$. Since the latter is fixed, optimization only influences $L$. Yet, recall that the volume term in the change-of-variables objective *rewards* increasing $f$'s derivatives and thus $L$. While we have given an upper bound and therefore cannot say that increasing $L$ will necessarily decrease concentration in latent space, it is for certain that leaving $L$ unconstrained does not directly pressure the $f(\boldsymbol{x})$ evaluations to concentrate.

Previous work (Dinh et al., 2015; 2017; Kingma & Dhariwal, 2018) has almost exclusively used a factorized zero-mean Gaussian as the latent distribution, and therefore we examine this case in particular. The log-mode can be expressed as $-D/2 \cdot \log 2\pi\sigma_z^2$, making the likelihood bound

$$\log \mathrm{N}(f(\boldsymbol{x}; \boldsymbol{\phi}); \mathbf{0}, \sigma_z^2 \mathbb{I}) + \log \left|\frac{\partial \boldsymbol{f}_\phi}{\partial \boldsymbol{x}}\right| \leq \frac{-D}{2} \log 2\pi\sigma_z^2 + D \log L. \tag{8}$$

We see that both terms scale with $D$ although in different directions, with the contribution of the $z$-distribution becoming more negative and the volume term's becoming more positive. We performed a simulation to demonstrate this behavior on the two moons data set, which is shown in Figure 7 (a). We replicated the original two dimensions to create data sets of dimensionality of up to 100. The results are shown in Figure 7 (b). The empirical values of the two terms are shown by the solid lines, and indeed, we see they exhibit the expected diverging behavior as dimensionality increases.

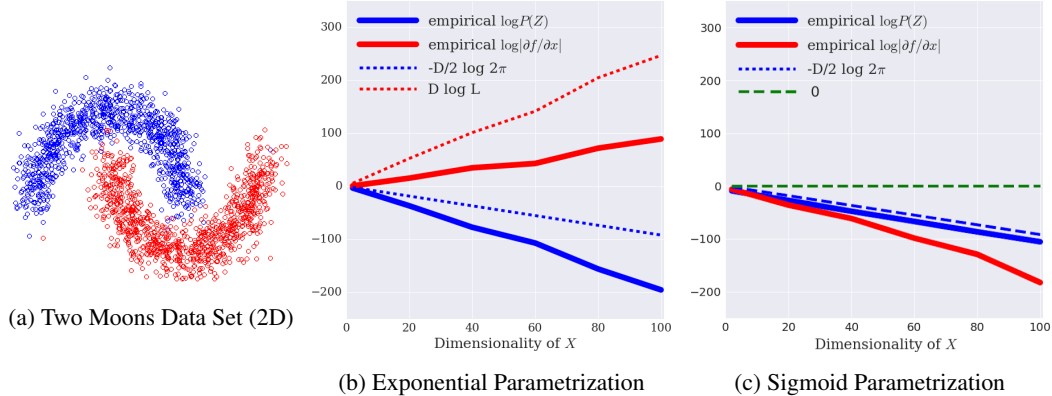

(a) Two Moons Data Set (2D)

(b) Exponential Parametrization

(c) Sigmoid Parametrization

Figure 7: *Limiting Bounds.* We trained an RNVP transformation on two moons data sets—which is shown in (a) for 2 dimensions—of increasing dimensionality, tracking the empirical value of each term against the upper bounds. Subfigure (b) shows Glow with an $\exp$ parametrization for the scales and (c) shows Glow with a sigmoid parametrization.

## D  GLOW WITH SIGMOID PARAMETRIZATION

Upon reading the open source implementation of Glow,[4] we found that Kingma & Dhariwal (2018) in practice parametrize the scaling factor as $\texttt{sigmoid}(s(\boldsymbol{x}_{d:}; \boldsymbol{\phi}_s))$ instead of $\exp\{s(\boldsymbol{x}_{d:}; \boldsymbol{\phi}_s)\}$. This

---

[4]https://github.com/openai/glow/blob/master/model.py#L376

choice allows the volume only to decrease and thus results in the volume term being bounded as (ignoring the convolutional transforms)

$$\log\left|\frac{\partial \boldsymbol{f}_\phi}{\partial \boldsymbol{x}}\right| = \sum_{f=1}^{F}\sum_{j=1}^{d_f}\log \texttt{sigmoid}(s_{f,j}(\boldsymbol{x}_{d_f:};\phi_s)) \leq FD\log 1 = 0 \qquad (9)$$

where $f$ indexes the flows and $d_f$ the dimensionality at flow $f$. Interestingly, this parametrization has a fixed upper bound of zero, removing the dependence on $D$ found in Equation 8. We demonstrate the change in behavior introduced by the alternate parametrization via the same two moon simulation. The only difference is that the RNVP transforms use a sigmoid parametrizations for the scaling operation. See Figure 7 (c) for the results: we see that now both change-of-variable terms are oriented downward as dimensionality grows. We conjecture this parametrization helps condition the log-likelihood, limiting the volume term's influence, when training the large models ($\sim$ 90 flows) used by Kingma & Dhariwal (2018). However, it does not fix the out-of-distribution over-confidence we report in Section 3.

## E    Constant and Random Inputs

| Data Set | Avg. Bits Per Dimension | Data Set | Avg. Bits Per Dimension |
|---|---|---|---|
| *Glow Trained on FashionMNIST* | | *Glow Trained on CIFAR-10* | |
| Random | 8.686 | Random | 15.773 |
| Constant (0) | **0.339** | Constant (128) | **0.589** |

Figure 8: *Random and constant images.* Log-likelihood (expressed in bits per dimension) of random and constant inputs calculated from NVP-Glow for models trained on FashionMNIST (left) and CIFAR-10 (right).

## F    Ensembling Glows

The likelihood function technically measures how likely the parameters are under the data (and not how likely the data is under the model), and perhaps a better quantity would be the posterior predictive distribution $p(\boldsymbol{x}_{test}|\boldsymbol{x}_{train}) = \frac{1}{M}\sum_m p(\boldsymbol{x}_{test}|\boldsymbol{\theta}_m)$ where we draw samples from posterior distribution $\boldsymbol{\theta}_m \sim p(\boldsymbol{\theta}|\boldsymbol{x}_{train})$. Intuitively, it seems that such an integration would be more robust than a single maximum likelihood point estimate. As a crude approximation to Bayesian inference, we tried averaging over ensembles of generative models since Lakshminarayanan et al. (2017) showed that ensembles of discriminative models are robust to out-of-distribution inputs. We compute an "ensemble predictive distribution" as $p(\boldsymbol{x}) = \frac{1}{M}\sum_m p(\boldsymbol{x};\boldsymbol{\theta}_m)$, where $m$ indexes over models. However, as Figure 9 shows, ensembles did not significantly change the relative difference between in-distribution (CIFAR-10, black and blue) and out-of-distribution (SVHN, red).

## G    Derivation of CV-Glow's Likelihood Difference

We start with Equation 5:

$$\frac{1}{2}\operatorname{Tr}\left\{\left[\nabla^2_{\boldsymbol{x}_0}\log p_z(f(\boldsymbol{x}_0;\phi)) + \nabla^2_{\boldsymbol{x}_0}\log\left|\frac{\partial \boldsymbol{f}_\phi}{\partial \boldsymbol{x}_0}\right|\right](\boldsymbol{\Sigma}_q - \boldsymbol{\Sigma}_{p^*})\right\}.$$

The volume element for CV-Glow does not depend on $\boldsymbol{x}_0$ and therefore drops from the equation:

$$\nabla^2_{\boldsymbol{x}_0}\log\left|\frac{\partial \boldsymbol{f}_\phi}{\partial \boldsymbol{x}_0}\right| = \nabla^2_{\boldsymbol{x}_0}HW\sum_k\log|\boldsymbol{U}_k| = 0 \qquad (10)$$

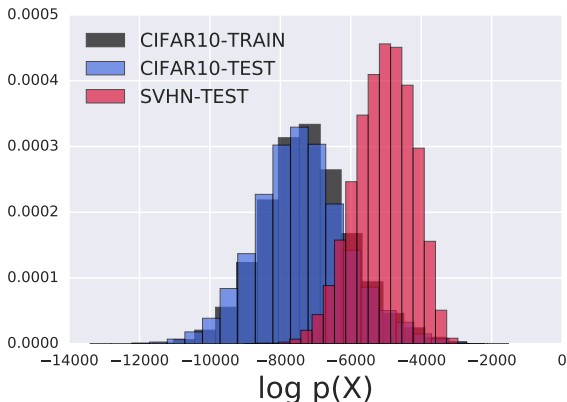

Figure 9: *Ensemble of Glows.* The plot above shows a histogram of log-likelihoods computed using an ensemble of Glow models trained on CIFAR-10, tested on SVHN. Ensembles were not found to be robust against this phenomenon.

where $\boldsymbol{U}_k$ denotes the $k$th $1 \times 1$-convolution's kernel. Moving on to the first term, the log probability under the latent distribution, we have:

$$
\begin{aligned}
\nabla^2_{\boldsymbol{x}_0} \log p(f(\boldsymbol{x}_0); \boldsymbol{\psi}) &= \nabla^2_{\boldsymbol{x}_0} \left\{ \frac{-1}{2\sigma^2_{\boldsymbol{\psi}}} \|f(\boldsymbol{x}_0)\|^2_2 - \frac{D}{2} \log 2\pi\sigma^2_{\boldsymbol{\psi}} \right\} \\
&= \nabla_{\boldsymbol{x}_0} \left\{ \frac{-1}{\sigma^2_{\boldsymbol{\psi}}} \left( \sum_d f_d(\boldsymbol{x}_0) \right) \nabla_{\boldsymbol{x}_0} f(\boldsymbol{x}_0) \right\} \\
&= \frac{-1}{\sigma^2_{\boldsymbol{\psi}}} \left[ \nabla_{\boldsymbol{x}_0} f(\boldsymbol{x}_0)(\nabla_{\boldsymbol{x}_0} f(\boldsymbol{x}_0))^T + \left( \sum_d f_d(\boldsymbol{x}_0) \right) \nabla^2_{\boldsymbol{x}_0} f(\boldsymbol{x}_0) \right].
\end{aligned}
\tag{11}
$$

Since $f$ is comprised of translation operations and $1 \times 1$ convolutions, its partial derivatives involve just the latter (as the former are all ones), and therefore we have the partial derivatives:

$$
\frac{\partial f_{h,w,c}(\boldsymbol{x}_0)}{\partial x_{h,w,c}} = \prod_{k=1}^{K} \sum_{j=1}^{C_k} u_{k,c,j}, \qquad \frac{\partial^2 f_{h,w,c}(\boldsymbol{x}_0)}{\partial x^2_{h,w,c}} = 0
\tag{12}
$$

where $h$ and $w$ index the input spatial dimensions, $c$ the input channel dimensions, $k$ the series of flows, and $j$ the column dimensions of the $C_k \times C_k$-sized convolutional kernel $\boldsymbol{U}_k$. The diagonal elements of $\nabla_{\boldsymbol{x}_0} f(\boldsymbol{x}_0)(\nabla_{\boldsymbol{x}_0} f(\boldsymbol{x}_0))^T$ are then $(\prod_{k=1}^{K} \sum_{j=1}^{C_k} u_{k,c,j})^2$, and the diagonal element of $\nabla^2_{\boldsymbol{x}_0} f(\boldsymbol{x}_0)$ are all zero.

Then returning to the full equation, for the constant-volume Glow model we have:

$$
\begin{aligned}
&\frac{1}{2} \operatorname{Tr} \left\{ \left[ \nabla^2_{\boldsymbol{x}_0} \log p(f(\boldsymbol{x}_0)) + \nabla^2_{\boldsymbol{x}_0} \log \left| \frac{\partial \boldsymbol{f}}{\partial \boldsymbol{x}_0} \right| \right] (\boldsymbol{\Sigma}_q - \boldsymbol{\Sigma}_{p^*}) \right\} \\
&= \frac{1}{2} \operatorname{Tr} \left\{ \left[ \nabla^2_{\boldsymbol{x}_0} \log p(f(\boldsymbol{x}_0)) \right] (\boldsymbol{\Sigma}_q - \boldsymbol{\Sigma}_{p^*}) \right\} \\
&= \frac{-1}{2\sigma^2_{\boldsymbol{\psi}}} \operatorname{Tr} \left\{ \left[ \nabla_{\boldsymbol{x}_0} f(\boldsymbol{x}_0)(\nabla_{\boldsymbol{x}_0} f(\boldsymbol{x}_0))^T + \left( \sum_d f_d(\boldsymbol{x}_0) \right) \nabla^2_{\boldsymbol{x}_0} f(\boldsymbol{x}_0) \right] (\boldsymbol{\Sigma}_q - \boldsymbol{\Sigma}_{p^*}) \right\} \\
&= \frac{-1}{2\sigma^2_{\boldsymbol{\psi}}} \sum_{l,m} \left\{ \left[ \nabla_{\boldsymbol{x}_0} f(\boldsymbol{x}_0)(\nabla_{\boldsymbol{x}_0} f(\boldsymbol{x}_0))^T + \left( \sum_d f_d(\boldsymbol{x}_0) \right) \nabla^2_{\boldsymbol{x}_0} f(\boldsymbol{x}_0) \right] \odot (\boldsymbol{\Sigma}_q - \boldsymbol{\Sigma}_{p^*}) \right\}_{l,m}.
\end{aligned}
\tag{13}
$$

Lastly, we assume that both $\Sigma_q$ and $\Sigma_{p^*}$ are diagonal and thus the element-wise multiplication with $\nabla^2_{\boldsymbol{x}_0} \log p(f(\boldsymbol{x}_0))$ collects only its diagonal elements:

$$
\begin{aligned}
& \frac{-1}{2\sigma^2_{\boldsymbol{\psi}}} \sum_{l,m} \left\{ \left[ \nabla_{\boldsymbol{x}_0} f(\boldsymbol{x}_0)(\nabla_{\boldsymbol{x}_0} f(\boldsymbol{x}_0))^T + \left( \sum_d f_d(\boldsymbol{x}_0) \right) \nabla^2_{\boldsymbol{x}_0} f(\boldsymbol{x}_0) \right] \odot (\Sigma_q - \Sigma_{p^*}) \right\}_{l,m} \\
& = \frac{-1}{2\sigma^2_{\boldsymbol{\psi}}} \sum_h^H \sum_w^W \sum_c^C \left( \prod_{k=1}^K \sum_{j=1}^{C_k} u_{k,c,j} \right)^2 (\sigma^2_{q,h,w,c} - \sigma^2_{p^*,h,w,c}) \\
& = \frac{-1}{2\sigma^2_{\boldsymbol{\psi}}} \sum_c^C \left( \prod_{k=1}^K \sum_{j=1}^{C_k} u_{k,c,j} \right)^2 \sum_h^H \sum_w^W (\sigma^2_{q,h,w,c} - \sigma^2_{p^*,h,w,c})
\end{aligned}
\tag{14}
$$

where we arrived at the last line by rearranging the sum to collect the shared channel terms.

## H  HISTOGRAM OF DATA STATISTICS

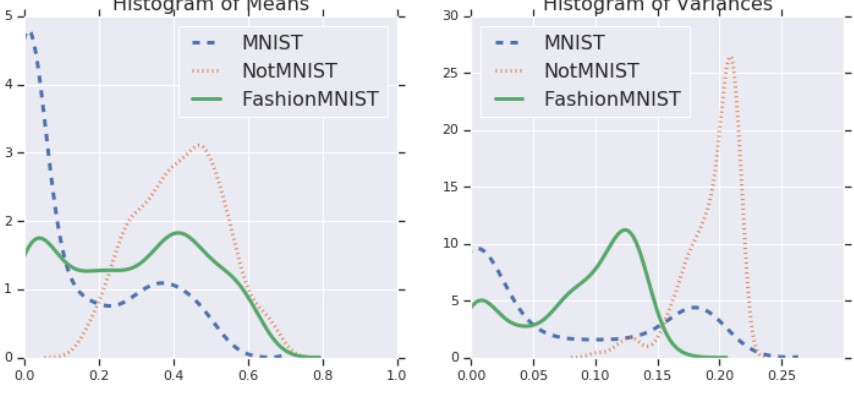

(a) Datasets of $28 \times 28 \times 1$ images: MNIST, FashionMNIST and NotMNIST.

Figure 10: Data statistics: Histogram of per-dimensional mean, computed as $\mu_d = \frac{1}{N} \sum_{n=1}^N x_{nd}$, and per-dimensional variance, computed as $\sigma^2_d = \frac{1}{N-1} \sum_{n=1}^N (x_{nd} - \mu_d)^2$. Note that pixels are converted from 0-255 scale to 0-1 scale by diving by 256. See Figure 5a for results on datasets of $32 \times 32 \times 3$ images: SVHN, CIFAR-10, CIFAR-100, CelebA and ImageNet.

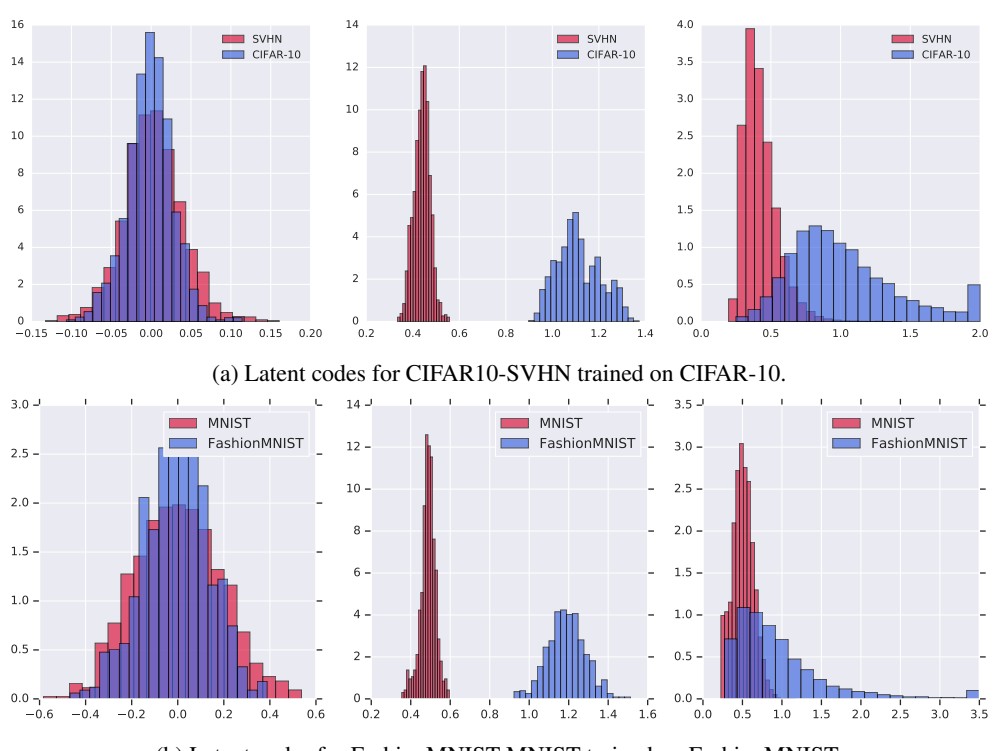

(a) Latent codes for CIFAR10-SVHN trained on CIFAR-10.

(b) Latent codes for FashionMNIST-MNIST trained on FashionMNIST.

Figure 11: Analysis of codes obtained using CV-Glow model. Histogram of means (left column) $\mu_d = \frac{1}{N} \sum_{n=1}^{N} z_{nd}$, standard deviation (middle column) $\sigma_d = \sqrt{\frac{1}{N-1} \sum_{n=1}^{N} (z_{nd} - \mu_d)^2}$ and norms normalized by $\sqrt{D}$ (right column) computed as $\frac{|z_n|}{\sqrt{D}} = \sqrt{\frac{1}{D} \sum_d z_{nd}^2}$.

# I  RESULTS ILLUSTRATING EFFECT OF GRAYING ON CODES

Figure 12 shows the effect of graying on codes.

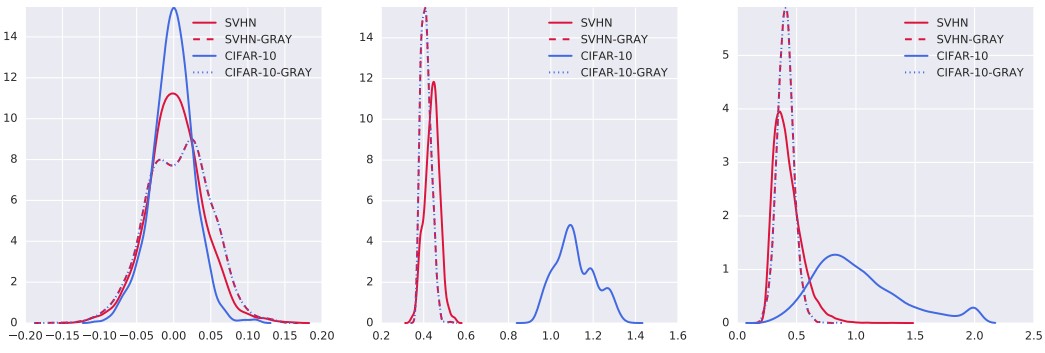

(a) CV-Glow trained on CIFAR-10: Effect of graying on CIFAR-10 and SVHN codes

Figure 12: Effect of graying on codes. Left (mean), middle (standard deviation) and norm (right).

# J  SAMPLES

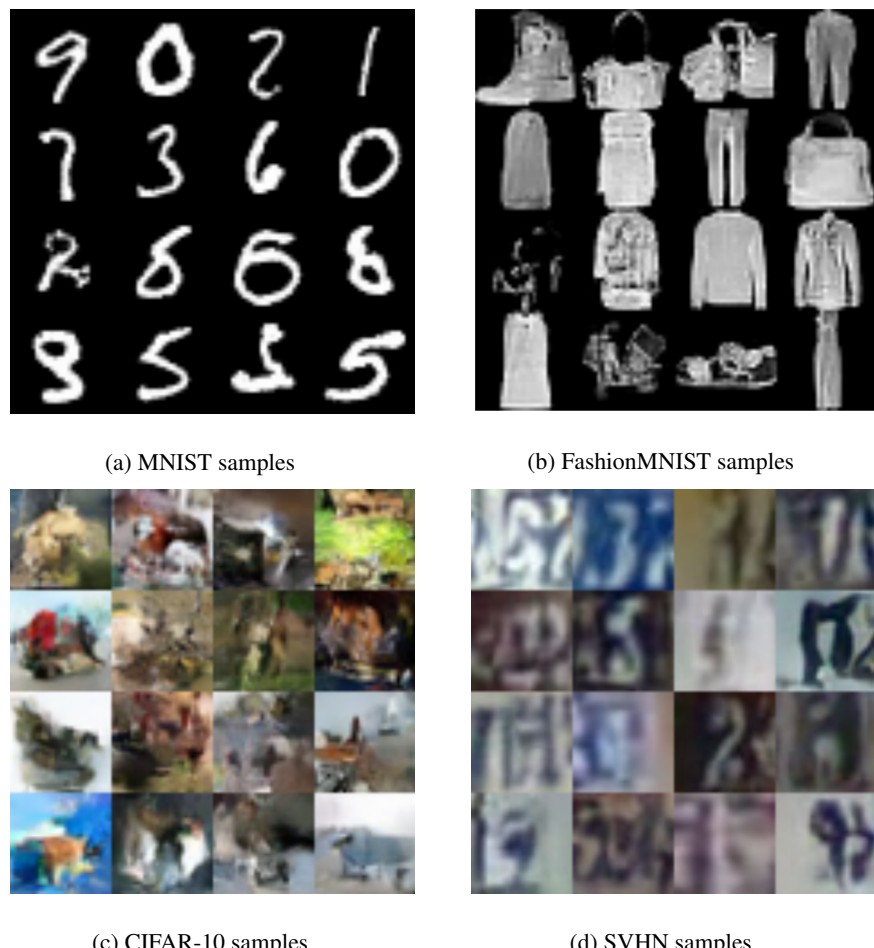

(a) MNIST samples

(b) FashionMNIST samples

(c) CIFAR-10 samples

(d) SVHN samples

Figure 13: *Samples.* Samples from CV-Glow models used for analysis.

