# OpenReview forum: "Do Deep Generative Models Know What They Don't Know? "
_ICLR.cc/2019/Conference_

### Official Review · AnonReviewer3 · 2018-10-29
**Very interesting finding; insufficient empirical analysis, theory with approximations too bold**

**Rating:** 7
**Confidence:** 3

**Review:**

Pros:
- The finding that SVHN has larger likelihood than CIFAR according to networks is interesting.
- The empirical and theoretical analyses are clear, seem thorough, and make sense.
- Section 5 can provide some insight when the model is too rigid and too log-concave (e.g. Gaussian).
Cons:
- The premises of the analyses are not very convincing, limiting the significance of the paper.
- In particular, Section 4 is a series of empirical analyses, based on one dataset pair. In 3/4 of the pairs the author tried, this phenomenon is not there. Whether the findings generalize to other situations where the phenomenon appears is uncertain.
- It is good that Section 5 has some theoretical analysis. But I personally find it very disturbing to base it on a 2nd order approximation of a probability density function of images when modeling something as intricate as models that generate images. At least this limitation should be pointed out in the paper.
- Some parts of the paper feel long-winded and aimless.

[Quality]
See above pros and cons.
A few less important disagreement I have with the paper:
- I don't think Glow necessarily is encouraged to increase sensitivity to perturbations. The bijection needs to map training images to a high-density region of the Gaussian, and that aspect would make the model think twice before making the volume term too large.
- Figure 6(a) clearly suggests that the data mean for SVHN and CIFAR are very different, instead of similar.

[Clarity]
In general, the paper is clear and easy to understand given enough reading time, but feels at times long-winded.
Section 2 background takes too much space.
Section 3 too much redundancy -- it just explains that SVHN has a higher likelihood when trained on CIFAR, and a few variations of the same experiment.
Section 4 seems to lack a high-level idea of what it want to prove -- the hypothesis around the volume term is dismissed shortly after, and it ultimately proves that we do not know what is the reason behind the high SVHN likelihood, making it look like a distracting side-experiment.
A few editorial issues:
- On page 4 footnote 2, as far as I know the paper did not define BPD.
- There are two lines of text between Fig. 4 and Fig. 5, which is confusing.

[Originality]
I am not an expert in this specific field (analyzing generative models), but I believe this analysis is novel.
However, there are papers empirically analyzing novelty detection using generative model -- should analyze or at least cite:
    Vít Škvára et al. Are generative deep models for novelty detection truly better?
    ^ at first glance, their AUROC is never under 0.5, indicating that this phenomenon did not appear in their experiments although a lot of inlier-novelty pairs are tried.
A part of the paper's contribution (section 5 conclusion) seem to overlap with others' work. The section concludes that if the second dataset has small variances, it will get higher likelihood. But this is too similar to the cited findings on page 6 (models assign high likelihood to constant images).

[Significance]
The paper has a very interesting finding; pointing out and in-depth analysis of negative results should benefit the community greatly.
However, only 1 dataset pair is experimented -- there should be more to ensure the findings generalize, since Sections 3 and 4 rely completely on empirical analysis. According to the conclusions of the paper, such dataset pairs should be easy to find -- just find a dataset that "lies within" another. Did you try e.g. CIFAR-100 train and CIFAR-10 test?
Section 5 is based on a 2nd order expansion on the $log p(x)$ given by a deep network -- I shouldn't be the judge of this, but from a realistic perspective this does not mean much.

---

> ### Author Response · Authors · 2018-11-26
> **Response to Reviewer #3**
>
> Thanks again, Reviewer #3, for your thought-provoking critique.  We respond to your other comments below.
>
> 1.  “In particular, Section 4 is a series of empirical analyses, based on one dataset pair….However, only 1 dataset pair is experimented -- there should be more to ensure the findings generalize, since Sections 3 and 4 rely completely on empirical analysis.”
>
> See general responses #1 and #3.
>
>
> 2.  “It is good that Section 5 has some theoretical analysis. But I personally find it very disturbing to base it on a 2nd order approximation of a probability density function of images when modeling something as intricate as models that generate images. At least this limitation should be pointed out in the paper….Section 5 is based on a 2nd order expansion on the $log p(x)$ given by a deep network -- I shouldn't be the judge of this, but from a realistic perspective this does not mean much.”
>
> See general response #2.  We emphasize that we are not trying to approximate the density function, only approximate the difference and characterize its sign.  Moreover, the special structure of CV-Glow makes these derivative-based approximations better behaved and more tractable than an expansion of a generic deep neural network.
>
>
> 3.  “Some parts of the paper feel long-winded and aimless….In general, the paper is clear and easy to understand given enough reading time, but feels at times long-winded.  Section 2 background takes too much space.  Section 3 too much redundancy -- it just explains that SVHN has a higher likelihood when trained on CIFAR, and a few variations of the same experiment.”
>
> We will attempt to make the writing more concise.  But we believe that most, if not all, of Section 2 is necessary in order to make the paper self-contained and accessible to someone who has never before seen invertible generative models.  While we are fastidious in our experimental description in Section 3, we think it is necessary since this is the foundational section of the paper.
>
>
> 4.  “I don't think Glow necessarily is encouraged to increase sensitivity to perturbations. The bijection needs to map training images to a high-density region of the Gaussian, and that aspect would make the model think twice before making the volume term too large.”
>
> We are not saying that the model will totally disregard the latent density and attempt to scale the input to very large or infinite values.  Our point is made in the context of volume term which is only one of the terms in the change-of-variable objective. The log volume term in the change-of-variable objective is maximizing the very quantity (the Jacobian’s diagonal terms) that the cited work on derivative-based regularization penalties has sought to minimize.  The maximization of the derivatives in the objective directly implies increased sensitivity to perturbations.
>
>
> 5.  “Figure 6(a) [Figure 5(a) in revised draft] clearly suggests that the data mean for SVHN and CIFAR are very different, instead of similar.”
>
> We are not sure how you are drawing this conclusion; perhaps from the scale of the x-axis? The histogram in Figure 6 (a) (original draft) has an x-axis covering the interval [0.4, 0.55], meaning the maximal difference between a mean in *any pair of dimensions* is 0.15.  Scaling back to pixel units, 0.15 * 255 = 38.25, meaning that 38.25 pixels is the maximum difference in means.  While this is not a difference of zero, we don’t see how you could say this “clearly suggests” that the means are “very different.”  In the latest draft, this figure---now Fig 5 (a)---has an x-axis that spans from 0-255.  Hopefully the overlap in the means in now conspicuous.
>
>
> 6.  “However, there are papers empirically analyzing novelty detection using generative model -- should analyze or at least cite: Vít Škvára et al. Are generative deep models for novelty detection truly better? at first glance, their AUROC is never under 0.5, indicating that this phenomenon did not appear in their experiments although a lot of inlier-novelty pairs are tried.”
>
> Thank you for pointing us to this work.  We cite it in the revised draft.  It looks like they test on UCI data sets of dimensionality less than 200, and therefore their results speak to a much different data regime than the one we are studying.
>
>
> 7.  “A part of the paper's contribution (section 5 conclusion) seem to overlap with others' work. The section concludes that if the second dataset has small variances, it will get higher likelihood. But this is too similar to the cited findings on page 6 (models assign high likelihood to constant images).”
>
> While we do also analyze constant images, we believe that our results for multiple data set pairs (FashionMNIST-MNIST, CIFAR10-SVHN, CelebA-SVHN, ImageNet-CIFAR10/CIFAR100/SVHN) and for multiple deep generative models (flow-based models, VAE, PixelCNN) is novel. Our conclusions are arrived at through focused experimentation and a novel analytical expression applied to CV-Glow.

---

> > ### Comment · AnonReviewer3 · 2018-11-28
> > **Comments on the rebuttal**
> >
> > Thank you for your response. The extra results are promising, which makes the paper quite stronger. Other questions are addressed well. Now I am mainly focused on these three issues:
> >
> > 2. Second order analysis, but only on the *sign* of the *difference* of two pdfs
> >
> > I would think that since x is an image, it would be hard to approximate a distribution with a mixture of a thousand Gaussians, let alone one Gaussian. Even if you are taking the difference of two pdfs, and taking the sign of the difference, a Gaussian would give you a hypersphere, not a large amounts of irregular-shaped blobs scattered through the image space.
> >
> > It IS indeed inevitable that when theoretically analyzing deep networks, we have to start somewhere easy, and log-quadratic pdfs are a valid starting point. All I am asking is that the paper warns its readers of this shortcoming at the beginning of the analysis.
> >
> > 4. Loss actively increasing volume term unlike prior work
> >
> > It does seem that way, but by the same argument I can claim that any loss function function has a L2 component in it: if your loss is f(theta), then you just write f(theta) = g(theta) + |theta|_2^2, where g(theta) = f(theta) - L2. My bold claim only makes sense if in fact all terms in g(theta) collectively does not do much on the L2. Unfortunately this is not the case in this paper.
> >
> > Specifically in this paper, the latent density term is the happiest if you make f nearly degenerate (everything maps to a tiny proximity of argmax_z{ p(z) }, for example), making the volume term nearly zero. And the volume term is needed to change this into something meaningful. The two terms strike a balance. So it is not right to claim f(x) encourages sensitivity if one term encourages it and another discourages it. -- Especially considering the experiment fixing the volume term did not make SVHN and CIFAR closer. A better way to describe this story is can be along the lines of "one of the terms encourages the sensitivity (but the other discourages it), and that term makes SVHN likelihood pretty high, so one may think this is the issue. But we tried and it's not working".
> >
> > 5. Are SVHN and CIFAR centers close?
> >
> > *Individually*, each dimension of the means is quite close, but remember that two mean vectors are close only if *everything* is close.  These are I assume 32x32=1024 feature space, so you would amplify the estimated 0.15 by 1024, making it 150 which is huge (actual value is probably smaller). Since this is used for the difference of two distributions approximated by log-quadratics, one should see the drop of the approximated density function when you move as far as to the mean of the other distribution. I am not convinced that it is small.

---

> > > ### Author Response · Authors · 2018-12-01
> > > **Re: Comments on the rebuttal**
> > >
> > > Thank you for your responses and continuing the discussion, Reviewer #3.  Our replies are below.
> > >
> > > 2.  "All I am asking is that the paper warns its readers of this shortcoming at the beginning of the analysis.":
> > >
> > > Fair point.  We will add a sentence at the beginning of Section 5 to make explicit that these expressions are approximations.
> > >
> > >
> > > 4.  We perfectly agree with your 'better description': "one of the terms encourages the sensitivity....But we tried and it's not working."  This is exactly what we wanted to convey in the draft, and we thought we clarified this point in our rebuttal by saying "Our point is made in the context of volume term which is only one of the terms in the change-of-variable objective."  We'll revise the draft to further emphasize our remarks pertain to the volume term only.
> > >
> > >
> > > 5.  "...making it 150 which is huge (actual value is probably smaller)"
> > >
> > > The difference is certainly much smaller.  It would be 150 only if the histograms were perfectly separated to each end of the x-axis in Figure 6 (a) of the original draft, which is not the case at all.  What metric / plot would convince you?  Some statistic of the dimension-wise means?

---

> > > > ### Comment · AnonReviewer3 · 2018-12-01
> > > > **Thank you for your feedback**
> > > >
> > > > And thank you for revising the text. My main concerns are addressed, and the issue #5 is pretty minor given the other assumption made in the analysis.
> > > >
> > > > I am not a statistics expert, if one wants to test whether two univariate Gaussians have different means or not, a student-t test can be used. In this case of multivariate Gaussians, a brief search suggests using its generalization, "Hotelling's two-sample t-squared statistics/test". In the end, one wants to compare the distance (considering different dimensions have different correlations, the Mahalanobis distance is better) between the two means, and compare its scale to the covariance matrices of both Gaussians.
> > > >
> > > > A rougher test is see if one Gaussian's mean lies inside the confidence interval of the other Gaussian. See multivariate normal distribution's confidence interval.
> > > >
> > > > In the case that the tests fail, one can see how much the test statistics are larger than e.g. the 95% quantile of the corresponding test distributions.

---

> > > > > ### Author Response · Authors · 2018-12-02
> > > > > **Re: Thank you for your feedback**
> > > > >
> > > > > Thank you for these suggestions, Reviewer #3.  We probably won't be able to add them in the next week---as many of us authors are traveling to / attending NeurIPS---but we will add them to the next iteration of the draft.

---

### Official Review · AnonReviewer2 · 2018-11-01
**Interesting example of density modelling shortcoming**

**Rating:** 6
**Confidence:** 4

**Review:**


This paper displays an occurrence of density models assigning higher likelihood to out-of-distribution inputs compared to the training distribution. Specifically, density models trained on CIFAR10 have higher likelihood on SVHN than CIFAR10. This is an interesting observation because the prevailing assumption is that density models can distinguish inliers from outliers. However, this phenomenon is not encountered when comparing MNIST and NotMNIST. The SVHN/CIFAR10 phenomenon has also been shown in concurrent work [1].

Given that you observed that SVHN has higher likelihood on all three model types (PixelCNN, VAE, Glow), why investigate a component specific to just flow-based models (the volume term)? It seems reasonable to suspect that the phenomenon may be due to a common cause in all three model types. For instance, the experiments seem to indicate that generalizing density estimation from CIFAR training set to CIFAR test set is likely challenging and thus the models underfit the true data distribution, resulting in the simpler dataset (SVHN) having higher likelihood.

Given the title of the paper, it would have been nice if this paper explored more than just MNIST vs NotMNIST and SVHN vs CIFAR10, so that the readers can gain a better feel for when generative models will be able to detect outliers. For instance, a scenario where the data statistics (pixel means and variances) are nearly equivalent for both datasets would be interesting. The second order analysis is good but it seems to come down to just a measure of the empirical variances of the datasets.

This paper is well written. I think the presentation of this density modelling shortcoming is a good contribution but leaves a bit to be desired.

[1] Choi, H. and Jang, E. Generative Ensembles for Robust Anomaly Detection. https://arxiv.org/abs/1810.01392


Pros:
- Interesting observation of density modelling shortcoming
- Clear presentation

Cons:
- Lack of a strong explanation for the results or a solution to the problem
- Lack of an extensive exploration of datasets

---

> ### Author Response · Authors · 2018-11-26
> **Response to Reviewer #2**
>
> Thanks again, Reviewer #2, for your insightful feedback.  We respond to your other comments below.
>
> 1.  “Why investigate a component specific to just flow-based models (the volume term)? It seems reasonable to suspect that the phenomenon may be due to a common cause in all three model types.”
>
> See general response #3.
>
>
> 2.  “For instance, the experiments seem to indicate that generalizing density estimation from CIFAR training set to CIFAR test set is likely challenging and thus the models underfit the true data distribution, resulting in the simpler dataset (SVHN) having higher likelihood.“
>
> We do not believe our models are necessarily underfit.  In fact, we found that Glow had a tendency to *overfit,* and that one must carefully set Glow’s l2 penalty and choose its scale parametrization (exp vs sigmoid, see Appendix D) in order to prevent it from doing so.  We thought this overfitting to the training data could be a reason for the phenomenon and therefore we tuned our implementations to have reasonable generalization.
>
>
> 3.  “It would have been nice if this paper explored more than just MNIST vs NotMNIST and SVHN vs CIFAR10, so that the readers can gain a better feel for when generative models will be able to detect outliers.  For instance, a scenario where the data statistics (pixel means and variances) are nearly equivalent for both datasets would be interesting.”
>
> See general response #1 in regards to data sets and additional results.  Thank you for the suggestion of looking at data sets with similar statistics.  We do this, in a way, with our second order analysis and the ‘gray-ing’ experiment in Figure 5 (b) (formerly Figure 6 (b) in the original draft).  Gray CIFAR-10 (blue dotted line) nearly overlaps with original SVHN (red solid line) in terms of their log p(x) evaluations.  Figure 12 (formerly Figure 13) then shows the latent (empirical) distribution of the gray images, and we see that the gray CIFAR-10 latent variables nearly overlap with the SVHN latent variables.  This is to be expected though, given the overlapping p(x) histograms, since the probability assigned by CV-Glow (in comparison to other inputs) is fully determined by the position in latent space.
>
> 4.  “The second order analysis is good but it seems to come down to just a measure of the empirical variances of the datasets.”
>
> See general response #2.

---

> > ### Comment · AnonReviewer2 · 2018-11-27
> > **Figure 4 d)**
> >
> >
> > From Figure 4 d), we see that, due to the inductive bias of the model, SVHN has lower bpd.
> > If the model were trained further, would the bpd of the training set ever become lower than SVHN test?
> >
> > If yes, then doesn't this indicate that, due to early stopping, the models are underfitting the CIFAR test set? In other words, generalizing density estimation from CIFAR training set to CIFAR test set is challenging and thus the models underfit the CIFAR test set, resulting in the simpler dataset (SVHN) having higher likelihood due to the inductive bias of the model. So possibly, given more data or a better inductive bias, this problem would go away?
> > If no, then it seems that the model is not complex enough since it is unable to obtain a lower bpd on CIFAR train compared to SVHN.
> >
> > Have you tested this? What are your thoughts?

---

> > > ### Author Response · Authors · 2018-11-27
> > > **Re: Figure 4 d)**
> > >
> > > No, the BPD never becomes lower for CIFAR-10 than for SVHN under any setting of the training time, optimization strategy, regularization type / strength, and model size that we tried.  It depends on what you mean by ’not complex enough.’  We achieve sampling and BDP numbers on par with SOTA so we don’t think that the explanation is simply to use a bigger model.  In fact, the Glow model trained by the authors of “Generative Models for Robust Anomaly Detection” (https://openreview.net/forum?id=B1e8CsRctX) is as large as Kingma & Dhariwal's (2018), and they report the same phenomenon.  If by ’not complex enough’ you mean that Glow could possibly be generally improved to better represent the training density, then sure, perhaps some innovation applied to Glow could make the model richer and fix the issue.  We do not believe such an innovation is trivial though, given how persistent the phenomenon is across hyperparameters and when ensembling (Appendix F).

---

### Official Review · AnonReviewer1 · 2018-11-03
**Interesting work and analysis**

**Rating:** 7
**Confidence:** 4

**Review:**

I really enjoyed reading the paper! The exposition is clear with interesting observations, and most importantly, the authors walk the extra mile in doing a theoretical analysis of the observed phenomena.

Questions for the authors:
1. (Also AREA CHAIR NOTE): Another parallel submission to ICLR titled “Generative Ensembles for Robust Anomaly Detection” makes similar observations and seemed to suggest that ensembling can help counter the observed CIFAR/SVHN phenomena unlike what we see in Figure 10. Their criteria also accounts for the variance in model log-likelihoods and is hence slightly different.
2. Even though Figure 2b shows that SVHN test likelihoods are higher than CIFAR test likelihoods, the overlap in the histograms of CIFAR-train and CIFAR-test is much higher than the overlap in CIFAR-train and SVHN-test. If we define both maximum and minimum thresholds based on the CIFAR-train histogram, it seems like one could detect most SVHN samples just by the virtue that there likelihoods are much higher than even the max threshold determined by the CIFAR-train histogram?
3. Why does the constant image (all zeros) in Figure 9 (appendix) have such a high likelihood? It’s mean (=0 trivially) is clearly different from the means of the CIFAR-10 images (Figure 6a) so the second order analysis of Section 5 doesn’t seem applicable.
4. How much of this phenomena do you think is characteristic for images specifically? Would be interesting to test anomaly detection using deep generative models trained on modalities other than images.
5. One of the anonymous comments on OpenReview is very interesting: samples from a CIFAR model look nothing like SVHN. This seems to call the validity of the anomalous into question. Curious what the authors have to say about this.

Minor nitpick: There seems to be some space crunching going on via Latex margin and spacing hacks that the authors should ideally avoid :)

---

> ### Author Response · Authors · 2018-11-26
> **Response to Reviewer #1**
>
> Thanks again, Reviewer #1, for your thoughtful comments.  We respond to your other comments below.
>
> 1.  “It seems like one could detect most SVHN samples just by the virtue that there likelihoods are much higher than even the max threshold determined by the CIFAR-train histogram?”
>
> This is an interesting idea, but we are not sure it is applicable.  If one looks closely at Figure 2 (b), there are still blue and black histogram bars (denoting CIFAR-10 train and test instances) covering the entirety of SVHN’s support (red bars).
>
>
> 2.  “[The constant input]’s mean (=0 trivially) is clearly different from the means of the CIFAR-10 images (Figure 6a) so the second order analysis of Section 5 doesn’t seem applicable.”
>
> See general response #2.
>
>
> 3.  “How much of this phenomena do you think is characteristic for images specifically? Would be interesting to test anomaly detection using deep generative models trained on modalities other than images.”
>
> We have not tested non-image data, since images are the primary focus of work on generative models, but this is an interesting area for future work.
>
>
> 4.  “Samples from a CIFAR model look nothing like SVHN. This seems to call the validity of the anomalous into question. Curious what the authors have to say about this.”
>
> This is a very good point.  See our response to Shengyang Sun’s comment below.  We see think this phenomenon has to do with concentration of measure and typical sets, but we do not yet have a rigorous explanation.
>
>
> 5.  “There seems to be some space crunching going on via Latex margin and spacing hacks that the authors should ideally avoid :)”
>
> We have fixed the spacing in the latest draft :)

---

> ### Public Comment · (anonymous) · 2018-11-28
> **Density Estimation Observation Appears Elsewhere**
>
> > 1. (Also AREA CHAIR NOTE): Another parallel submission to ICLR titled “Generative Ensembles for Robust Anomaly Detection” makes similar observations and seemed to suggest that ensembling can help counter the observed CIFAR/SVHN phenomena unlike what we see in Figure 10.
>
> The parallel submission called Deep Anomaly Detection with Outlier Exposure also makes the observation that SVHN examples have higher likelihood than CIFAR-10 examples, and they also propose a way to correct this behavior. This is in Section 4.4 of https://openreview.net/pdf?id=HyxCxhRcY7
> The results also suggest that SVHN results are one of the worst-cases for density estimators; density estimators are not as bad on many other datasets.

---

> > ### Author Response · Authors · 2018-12-06
> > **Re: Density Estimation Observation Appears Elsewhere**
> >
> > (Apologies for late response, we missed this earlier)
> >
> > Thanks for pointing us to your work.  We will incorporate it into our discussion of related work.

---

### Public Comment · (anonymous) · 2018-10-24
**Measurement and distribution**

Thanks very much for the excellent work. It is very interesting to see the distribution from this perspectives. I took a look on the paper Theis2016, it seems besides BPD, KLD, MMD, JSD are considered, is it possible that CIFAR10 and SVHN can be different based on these three measurement?

This also reminds me of domain shift problem, which aims to align p(x,y), can I understand in this way that although in data space, CIFAR and SVHN are similar (in term of the BPD number), however, in semantic level (y), they are still large gap between this two?

Thanks again for the excellent work~~

---

> ### Author Response · Authors · 2018-11-05
> **Re: Measurement and distribution**
>
> Thanks for your questions, comments, and compliments.  As for considering other divergences / discrepancies, indeed using these for either parameter estimation or evaluation could lead to different results.  It is an area of future work.  Given the prevalence of fitting models via maximum likelihood (KLD[p_empirical || p_model]), we thought reporting the result for just this divergence a worthy contribution.
>
> As for your second question, we’re not certain we completely understand your point.  Can you clarify a bit more, please?  A perceived mismatch between distance in pixel space vs semantic space may be due to natural images having a common global structure.  The models then extract mostly the shared structure and not the details that we visually cue upon.

---

### Public Comment · ~Shengyang_Sun4 · 2018-10-27
**Image Samples**

Thank you for this interesting work.

It is astonishing that a well-trained CIFAR10 model assigns larger log-likelihood to the SVHN datasets.

What confuses me is that why the samples from such models won't generate SVHN-like images. According to your derivation, the SVHN variances is only marginally smaller than CIFAR10 variances, therefore it is probably not due to that SVHN-like figures live in a much smaller subspace that are unlikely to sample from.

---

> ### Author Response · Authors · 2018-11-07
> **Re: Image Samples**
>
> Thank you for your comment, Shengyang.  This is a good point and something we were a bit puzzled by as well.  Our current hypothesis is that the SVHN samples do not fall within the model’s typical set.  To elaborate, in high dimensions samples at or very near to the mode are unlikely.  See the high-dimensional Gaussian example discussed here: https://www.inference.vc/high-dimensional-gaussian-distributions-are-soap-bubble/  While you are correct in that the variances in data space are not drastically different, the variances of each data set’s latent variables (Figure 12, top column, middle) are well separated, with SVHN’s variance being much smaller.  Thus the distribution in latent space may be a better way to characterize the model’s typical set as samples are first drawn in latent space and then passed to the inverse function.

---

### Author Response · Authors · 2018-11-26
**General Rebuttal (1/2)**

Thank you, reviewers, for your fair and helpful comments.  We’ve provided a general response below that addresses concerns common to multiple reviewers.  We’ll also respond to reviewers individually regarding issues particular to their review.

1.  Limited Number of Data Sets [R2, R3]:  We have now added additional results to Section 3 (Figures 1 and 2) showing that the phenomenon (higher likelihood on non-train data) occurs for FashionMNIST (train) vs MNIST (test), CelebA (train) vs SVHN (test), ImageNet (train) vs CIFAR10/CIFAR100/SVHN (test).  Furthermore, we have included these data sets into our plot of the empirical means and variances (Section 5), showing that our second-order analysis and ‘sitting inside of’-conclusion agrees with these additional observations.

2.  Accuracy / Generality of Second-Order Analysis [R1, R2, R3]:  All reviewers bring up questions about the second-order analysis.  Starting with R1, they question how Equation 5 applies / can be interpreted for constant images.  To slightly correct R1’s statement, the constant image with high likelihood under the SVHN-trained model is x=128.  Normalizing by the number of pixels, i.e. 128/265=0.5, places this constant image almost in the exact center of the means plot in Figure 6 (a)---thus, the second-order analysis does apply.  Then turning to Equation 5 and plugging in the variance Var[\delta(128)]=0, we have:

E_q [log p(x)] - E_p* [log p(x)] \approx ½ * (negative number for CV-Glow) * (0 - Sigma_p*) >= 0.

Hence the second-order analysis still holds for the delta function located at 128 and agrees with the empirical result.  We will add this derivation to the appendix.

Moving on to R2, they state that the second-order analysis reduces to “just a measure of the empirical variances of the datasets.”  This is true and was done so purposefully.  CV-Glow is the only generative model that we know of that (i) has high-capacity and (ii) is amenable to the second-order analysis.  For all other models mentioned (VAE, PixelCNN, NVP-Glow), the second-order equation depends on the second derivatives of the neural network w.r.t its input.  It’s hard, if not impossible, to say anything general about how these second derivatives behave across the input space, let alone across re-fittings of the model.  CV-Glow uniquely has second derivatives that simplify to a function of (i) the log-convexity of the latent distribution and (ii) the square of the 1x1 convolutional kernel’s parameters.  Since both of these terms have a constant sign, the interesting part of the equation does indeed boil down to “a measure of the empirical variances of the datasets.”  The complications introduced by the model have been taken out and what’s left is a function of the data statistics, which does allow for some general conclusions.  We will try to clarify this reasoning / motivation in the paper, as space permits.  Furthermore, the fact that our second-order analysis lead us to and agrees with the additional experiments (see general response #1) and the gray-ing attack (Figure 5 (b), formerly Figure 6 (b) in the original draft), we see this as evidence of its validity.

Lastly, we address R3’s comments that they “find it very disturbing to base [analysis] on a 2nd order approximation of a probability density function.”  We agree that trying to approximate a neural-network-based density with only a second-order representation is a tall order.  But this is not precisely what we are doing.  Rather, we are approximating *the difference* in density functions, and therefore we only care about *the sign* of the expression.  We believe the second-order expression is a useful representation for this.  Moreover, if we assume the data distributions have no cross-moments, then from Equation 11 we notice that the diagonal derivatives are zero for second-order and beyond, thus making the second-order expansion exact.  For these two reasons, we don’t believe our approximation is “disturbing.”  And since we are working with deep generative models, any analytical statements will require rather strong assumptions.

---

> ### Author Response · Authors · 2018-11-26
> **General Rebuttal (2/2)**
>
> 3.  Purpose / Direction of Section 4 [R2, R3]:  R2 asks “Why investigate a component specific to just flow-based models (the volume term)? It seems reasonable to suspect that the phenomenon may be due to a common cause in all three model types.”  While the phenomenon is common to multiple deep generative model classes, as Figure 3 shows, we found it very hard to analyze all three models simultaneously, on equal footing, due to their different structures and inference requirements.  For instance, how can we compare VAEs and PixelCNNs while controlling for the former’s approximate inference requirements?  How do we know any problems with densities / outlier detection aren’t due to a sub-optimal inference model or the variational approximation?  We thought we would make more headway by restricting the analysis to invertible models since they (i) admit exact likelihood calculations and (ii) have nice analytical properties stemming from the bijection constraint.  Having made this decision, we then thought the next natural step is to look at both terms in the change-of-variables objective---the density under p(z) and the volume term---to see if one of these in particular was the cause.  After seeing Figure 4 (c, d) (Figure 4 (a, b) in revised draft), we thought that the volume term is the culprit, which then lead to examination of constant-volume Glow (CV-Glow) (i.e. ‘constant volume’ across all inputs) as described on page 6.  While the volume term was a bit of a red herring, we thought the progression from {VAE, PixelCNN, NVP-Glow} → {NVP-Glow} → {CV-Glow} was a logical way to further examine the problem for an increasing tractable model class.
>
> Relatedly, R3 writes of Section 4: “Section 4 is a series of empirical analyses, based on one dataset pair….However, only 1 dataset pair is experimented -- there should be more to ensure the findings generalize, since Sections 3 and 4 rely completely on empirical analysis….Section 4 seems to lack a high-level idea of what it want to prove -- the hypothesis around the volume term is dismissed shortly after, and it ultimately proves that we do not know what is the reason behind the high SVHN likelihood, making it look like a distracting side-experiment.”  The purpose of focusing on just CIFAR-10 vs SVHN in Section 4 is to drill-down and isolate why the phenomenon is happening in this one particular case.  We think this is an appropriate approach, as we didn’t want to introduce too many experimental variables, as explained above.  Furthermore, the presence of this phenomenon for SVHN vs CIFAR-10 alone warrants investigation since those data sets are extremely popular in the ML community.  Yet, we have since added additional data sets (see general response #1) and hope the reviewer is now satisfied with this additional evidence of the phenomenon's prevalence.

---

### Author Response · Authors · 2018-11-26
**Revised Draft**

We have uploaded a revised draft in which we have attempted to incorporate the reviewers’ suggestions.  In particular, the new draft includes the following significant revisions:

1.  Additional Data Sets:  In Section 3 we now report results for Glow trained and tested on the following data sets (in addition to CIFAR-10 vs SVHN): FashionMNIST (train) vs MNIST (test), CelebA (train) vs SVHN (test), ImageNet (train) vs CIFAR10/CIFAR100/SVHN (test).  The phenomenon of interest (i.e. higher likelihood on out-of-distribution test data) is observed for all of these new pairs.  Furthermore, we include the empirical means and variances of these data sets in the analysis in Section 5 and show that they agree with our original draft’s conclusions.

2.  Related Work:  We discuss the Škvára et al. (2018) work (and other concurrent work) in Section 6, as suggested by Reviewer #3.

3.  Equation Spacing: We fix the spacing issue mentioned by Reviewer #1.

4.  Revised Plot of Empirical Means: Reviewer #3 had doubts about to what degree the data set means overlap.  We believe this doubt was due to the range of the x-axis in what was formerly Figure 6 (a)---now Figure 5 (a).  We have revised the figure to have range 0-255 (normalized to 0-1) and added the additional data sets.

5.  Removal of NotMNIST results: We have removed from the main text the NotMNIST vs MNIST experiment that was reported in the original draft.  However, the Appendix (most crucially Figures 8 and 13) still contains NotMNIST results and has not yet been updated with the new data sets.  We will fix this in the next draft.

---

> ### Public Comment · (anonymous) · 2018-12-12
> **on point 5 and asymmetric behaviour**
>
> could you please explain why the notmnist results were removed in the latest draft? I found these did illustrate well the issue this paper is trying to get across, albeit the asymmetric behaviour as reported in the appendix -- also, while on this, I'm also surprised that the official reviewers didn't ask more about this. Could you provide some thoughts on why reversing the train/test roles of data sets solves the pathological high test-likelihood issue? Thanks!

---

> > ### Author Response · Authors · 2018-12-18
> > **Re: on point 5 and asymmetric behaviour**
> >
> > Thanks for your comment and question.
> >
> > Per the reviewers' requests for more evidence of the phenomenon on additional data sets, we wanted to bolster the 'motivating observations' section with experiments that better exhibit the curious out-of-distribution behavior.  We found that the FashionMNIST-vs-MNIST pair illustrated the phenomenon better (i.e. larger BPD gap) than the NotMNIST-vs-MNIST pair and hence we replaced those results in the main text.   We will also add the corresponding plots to Appendix B showing the asymmetric behavior for this pair as well (due to time constraints, we couldn't update all of the figures in the Appendix during the rebuttal period).  This was the only reason for the switch.  If you think the NotMNIST-vs-MNIST experiment is more interesting for some other reason, please do let us know your thoughts.
> >
> > We wouldn't claim the asymmetry "solves" the issue since (i) even for models trained on SVHN, there could be other datasets that lead to higher likelihood and (ii) it does not immediately reveal a procedure to correct the CIFAR10-vs-SVHN (or similar) issue.  The second-order analysis in Section 5 is still our best explanation for the asymmetric behavior.  That is, the interaction between the model curvature and the data set variance leads to the phenomenon, and when the sign of the difference in variances is switched (which occurs when the train and OOD sets are switched), then we expect the phenomenon behavior to flip as well.

---

### Meta-Review · Area_Chair1 · 2018-12-13
**Interesting empirical observation and analysis**

**Confidence:** 4
**Recommendation:** Accept (Poster)

**Metareview:**

This paper makes the intriguing observation that a density model trained on CIFAR10 has higher likelihood on SVHN than CIFAR10, i.e., it assigns higher probability to inputs that are out of the training distribution. This phenomenon is also shown to occur for several other dataset pairs. This finding is surprising and interesting, and the exposition is generally clear. The authors provide empirical and theoretical analysis, although based on rather strong assumptions. Overall, there's consensus among the reviewers that the paper would make a valuable contribution to the proceedings, and should therefore be accepted for publication.